e-science

open science, research policy, cumulative advantage, equity

**Author for correspondence:**
Tony Ross-Hellauer
e-mail: tross@know-center.at

# Dynamics of cumulative advantage and threats to equity in open science: a scoping review

Tony Ross-Hellauer[1,2], Stefan Reichmann[2], Nicki Lisa Cole[1,2], Angela Fessl[1,2], Thomas Klebel[1] and Nancy Pontika[3]

[1]Know-Center GmbH, Graz, Austria
[2]Open and Reproducible Research Group, Graz University of Technology, Inffeldgasse 13, 8010 Graz, Austria
[3]The Open University, Milton Keynes, UK

TR-H, 0000-0003-4470-7027; NLC, 0000-0002-6034-533X

Open Science holds the promise to make scientific endeavours more inclusive, participatory, understandable, accessible and re-usable for large audiences. However, making processes open will not *per se* drive wide reuse or participation unless also accompanied by the capacity (in terms of knowledge, skills, financial resources, technological readiness and motivation) to do so. These capacities vary considerably across regions, institutions and demographics. Those advantaged by such factors will remain potentially privileged, putting Open Science's agenda of inclusivity at risk of propagating conditions of 'cumulative advantage'. With this paper, we systematically scope existing research addressing the question: 'What evidence and discourse exists in the literature about the ways in which dynamics and structures of inequality could persist or be exacerbated in the transition to Open Science, across disciplines, regions and demographics?' Aiming to synthesize findings, identify gaps in the literature and inform future research and policy, our results identify threats to equity associated with all aspects of Open Science, including Open Access, Open and FAIR Data, Open Methods, Open Evaluation, Citizen Science, as well as its interfaces with society, industry and policy. Key threats include: stratifications of publishing due to the exclusionary nature of the author-pays model of Open Access; potential widening of the digital divide due to the infrastructure-dependent, highly situated nature of open data practices; risks of diminishing qualitative methodologies as 'reproducibility' becomes synonymous with quality; new risks of bias and exclusion in means of

transparent evaluation; and crucial asymmetries in the Open Science relationships with industry and the public, which privileges the former and fails to fully include the latter.

## 1. Introduction

Academia remains critically inequitable. The Global North dominates authorship and collaborative research networks, pushing the Global South to the periphery [1,2]. Even within richer regions, a fetish for the poorly defined goal of 'excellence' [3] breeds cumulative advantage in funding allocation for the highest-funded institutions [4]. At the level of individuals, early success shapes future success [5]. Women occupy relatively fewer higher positions, tend to achieve senior positions at a later age, are awarded less grant funding and have fewer 'high-impact publications' [6–9]. Lack of equity has been found to shut out participation in the scientific conversation and potentially reduce motivation, happiness and willingness to work, even among those who actually benefit [10]. These inequalities undoubtedly testify to broader societal imbalances but, as observed since the 1960s [11], dynamics of social mobility play out in academia in specific ways (cf. [12]).

Open Science[1] has been proposed at least in part as a corrective for some of these issues. Open Science has been defined as 'transparent and accessible knowledge that is shared and developed through collaborative networks' [13]. It is a varied movement to reform research through more transparent and participatory practices including Open Access to publications, research data sharing, opening research methods and processes, new means of transparent research evaluation and the re-orientation of research to be more inclusive of and responsive to the needs of society and industry [14]. Its motivations are diverse. Fernández Pinto [15] argues that Open Science can be variously seen, *inter alia*, as a culture, a goal, a movement, a set of policies, a project and a research strategy. Fecher & Friesike's [16] definition of Open Science is as an 'umbrella term encompassing a multitude of assumptions'. They identify five distinct 'schools of thought' reflecting the diverse motivations underpinning Open Science:

— *Infrastructure School:* Aims to create open platforms, tools and services to enable efficient and collaborative research.
— *Public School:* Aims to make science accessible to citizens and others beyond academia.
— *Measurement School:* Aims to develop alternative assessment systems for research.
— *Democratic School:* Aims to make knowledge freely available to everyone.
— *Pragmatic School:* Aims to make scientific processes more efficient, collaborative and open.

Social and epistemic justice are central to at least two of these motivations (Democratic and Public Schools), but important drivers of all. Equity has been a key aim of Open Science since its inception. The stirring language of the foundational 2002 Budapest Open Access Initiative, for example, claimed Open Access could share learning between rich and poor and 'lay the foundation for uniting humanity in a common intellectual conversation and quest for knowledge' [17]. Nielsen's seminal *Reinventing Discovery* devotes a chapter to the ways in which networked Open Science is 'democratizing' research [18]. More recently, 'increased equity' was listed as a 'key success factor' for Open Science by a stakeholder-driven study [19]. As Grahe *et al.* [20] say, 'Open science principles of openness and transparency provide opportunities to advance diversity, justice and sustainability by promoting diverse, just and sustainable outcomes'.

However, equity is one aim of Open Science among others, including increasing research quality and efficiency. Depending on definitions and priorities, these overlapping aims may conflict. What is more, these aims are necessarily refracted through the competing motivations of a myriad of actors (including researchers, research institutions, funders, governments, publishers). The equivocal nature of Open Science hence leaves room for interpretative flexibility in adoption and implementation, while its heavy political and economic implications mean that diverse and potentially conflicting motives are at play. Disconnects between expressed ideals and eventual policies and practices should be expected.

This is especially so since academia seems perniciously vulnerable to logics of 'cumulative advantage', as has been recognized at least since Merton proposed the existence of the 'Matthew

[1]In English, the word 'science' is taken to exclude the arts and humanities. Hence the term 'Open Science' is often taken to be exclusionary of these domains, and more inclusive terms like 'open scholarship' or 'open research' can be preferred by some. We here use the more common term 'Open Science', but this should be read as referring to research from all academic disciplines.

effect', whereby already successful scientists tend to receive disproportionately high recognition or rewards (e.g. reputation, resources, access to infrastructure) in comparison to their less-famous counterparts [21–23]. For Merton, 'systems of reward, allocation of resources and other elements of social selection thus operate to create and to maintain a class structure in science by providing a stratified distribution of chances among scientists for significant scientific work' [23]. Subsequent research identified the Matthew effect at work in research at the level of article citations [24], journals [25], institutions [26], departments [27] and countries [28], and along persistent fault lines of inequality like race [29] and gender [30]. It is at work across a range of scientific activities, including peer review [31], public engagement [32] and funding acquisition [33]. Although for Merton the Matthew effect was potentially detrimental in clustering resources and stifling innovation, he also saw it as a functional element aiding assessment of the credibility of sources, allocation of attention and recognizing outstanding contributions [31]. But while the Matthew effect in its various forms might be functional at a system level, it no doubt has the effect of advantaging and disadvantaging the contributions of individuals, as well as the individuals themselves, based on secondary attributes. Given the equity aim of Open Science, this is problematic *per se*.

Merton later broadened his thought to identify the Matthew effect as an example of cumulative advantage, whereby 'comparative advantages of trained capacity make for successive increments of structural location, and available resources make for advantage such that the gaps between the haves and the have-nots in science (as in other domains of social life) widen until dampened by countervailing processes' [23]. The lines delineating the Matthew effect and cumulative advantage are often blurred.[2] For our purposes, and to avoid confusion, in what follows we will prefer the broader term cumulative advantage, and define it along with DiPrete & Eirich [38] as 'a general mechanism for inequality across any temporal process … in which a favourable relative position becomes a resource that produces further relative gains'. These mechanisms are also closely related to what is referred to as preferential attachment in network theory, where power-law distributions are a result of the positionality and individual attributes of specific agents as nodes in a network shape possibilities for future accrual of resources within that network, such as larger nodes having more possibility for connection [39].

We hence understand Open Science as a diverse agenda to increase transparency, accessibility and participation in research, where equity is a commonly stated aim. We also, however, understand that various aspects of academia are particularly vulnerable to the logics of cumulative advantage. Bringing these threads together, we are led to ask whether Open Science is itself affected by such mechanisms, and whether they endanger the equity aim of Open Science.

As argued by Albornoz *et al.* [40], Open Science policies are situated within power imbalances and historical inequalities with respect to knowledge production (cf. [41]). Uncritical narratives of openness, therefore, may fail to address structural barriers in knowledge production and hence perpetuate the cumulative advantage of dominant groups and the knowledge they produced. Making processes open requires capacities (in terms of knowledge, skills, financial resources, political will, technological readiness and motivation) which vary across regions, institutions and demographics. In addition, persistent structural inequalities and social and cognitive biases will not be eliminated in an Open Science world. We must, therefore, ask how equitable is the implementation of Open Science across a range of stakeholder categories, in particular those at the peripheries? Might interventions in some cases actually deepen inequalities or be at conflict with wider Open Science goals? How do geographical, socio-economic, cultural and structural conditions lead to peripheral configurations in the Open Science landscape? What factors are at play and what can be done (at a policy level) to enhance uptake and contribution to the production of scientific knowledge by everybody?

With this paper, we aim to systematically scope existing research to answer the question: 'What evidence and discourse exists in the literature about the ways in which dynamics and structures of inequality could persist or be exacerbated in the transition to Open Science, across disciplines, regions and demographics?' Our scope includes all aspects of Open Science, including Open Access, Open Data, FAIR Data, Open Methods, Open Evaluation and Citizen Science, as well as its interactions with the interfaces between science and society and industry. Results are presented according to these dimensions. This will synthesize evidence and discourse, identify gaps in the literature and inform future research and policy. Given that the intention is to describe the general scope of the issues, no systematic quality appraisal of studies is carried out.

---

[2]Although sociologists may identify it solely as referring to Merton's original context of scholarly reputation and rewards, the Matthew effect has been taken up to describe phenomena of accumulation in areas as diverse as online markets [34], reading and literacy [35], sexual networks [36] and transitions to democracy [37].

This study uses the PRISMA framework [42] to align study selection with the research question and will follow the relevant aspects of the PRISMA Extension for Scoping Reviews to ensure thorough mapping, reporting and analysis of the literature. As Tricco *et al.* state, scoping reviews are useful to 'examine the extent (that is, size), range (variety) and nature (characteristics) of the evidence on a topic or question; determine the value of undertaking a systematic review; summarize findings from a body of knowledge that is heterogeneous in methods or discipline; or identify gaps in the literature to aid the planning and commissioning of future research'.

Since the many potential benefits of Open Science have been well-argued elsewhere [43–46], our presentation here necessarily focuses in greater depth on those areas where Open Science implementation potentially endangers the aim of greater equity in science. This emphasis should not be interpreted as signalling that the authors believe that the negatives outweigh the positives. Yet Open Science has now undoubtedly come of age, as mainstream policy in many regions and institutions, and must itself be open to critical and continued reflection upon the ways in which implementation may run counter to ideals. We believe such critique should be welcomed—above all by Open Science advocates—in order to re-orient implementation strategies and optimize outcomes wherever possible and desirable.

# 2. Methods

Methodologically, following identification of the above research question, the work has been structured according to the following four steps: identify relevant studies, select eligible studies, chart the data, collate and summarize the results.

## 2.1. Identifying relevant studies

A search was conducted for published and grey literature on the research area from January 2000 to the present, published in English. The authors first conducted a search of electronic databases (Scopus and Web of Science) on 23 December 2020 for citations and literature using the queries detailed below.

| |
| --- |
| Web of Science (All Databases)—1627 results |
| TOPIC: (('open science' OR 'science 2.0' OR 'Open Access' OR 'open peer review' OR 'altmetric*' OR 'alternative metric*' OR 'open data' OR 'reproducib*' OR 'FAIR Data' OR 'open innovation' OR 'citizen science') AND ('matthew effect*' OR 'cumulative advantage' OR 'inequ*' OR '*justice')) |
| Timespan: 2000–2020. Databases: WOS, BCI, BIOSIS, CCC, DIIDW, KJD, MEDLINE, RSCI, SCIELO |
| Search language = English |
| Scopus—1543 results |
| TITLE-ABS-KEY (('open science' OR 'science 2.0' OR 'Open Access' OR 'open peer review' OR 'altmetric*' OR 'alternative metric*' OR 'open data' OR 'reproducib*' OR 'FAIR Data' OR 'open innovation' OR 'citizen science') AND ('matthew effect*' OR 'cumulative advantage' OR 'inequ*')) PUBYEAR > 1999 AND (LIMIT-TO (LANGUAGE, 'English')) |

## 2.2. Selecting eligible studies

Searches yielded 3170 total results. Following manual deduplication, 2661 results remained for title/abstract screening, which was guided by the PRISMA framework, with specific eligibility criteria applied to ensure relevance for the study and its research questions. The selection process followed the recommendations in the Preferred Reporting Items for Systematic Reviews and Meta-Analyses Extension for Scoping Reviews (PRISMA-ScR) checklist and mapped using the PRISMA-P chart (figure 1). Web search engines and other sources were used to identify strongly relevant grey literature from bodies likely to have produced relevant grey literature reports such as research funders, research-performing organizations, academic publishers, student coalitions, OECD and UN. Finally, this was augmented by hand-searching references of the included studies and references (snow-balling). The following inclusion criteria were applied:

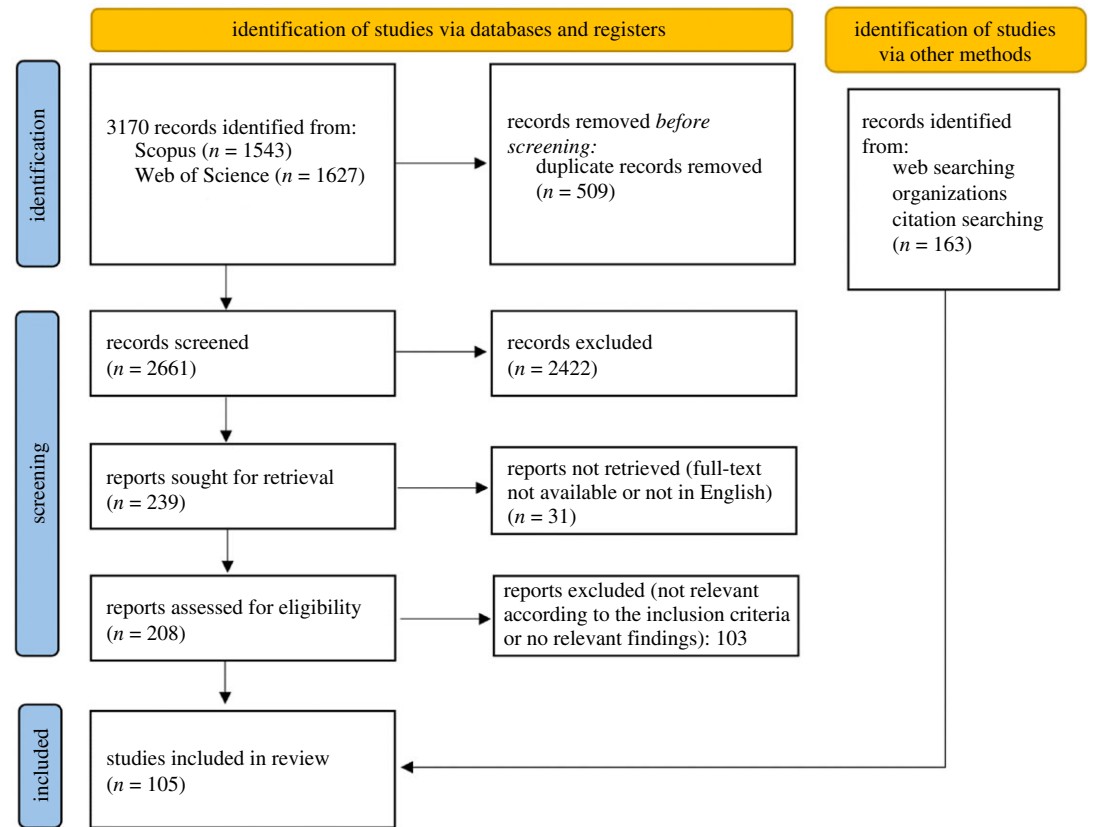

**Figure 1.** PRISMA diagram showing the literature searching and scoping process. Adapted from [42]. For more information, visit: http://www.prisma-statement.org/.

— Articles on potential effects in Open Science as they relate to the propagation of cumulative advantage.
— Conducted internationally or nationally.
— Published from 1 January 2000 until current.
— Available in English.
— Full-text could be obtained.
— Study is a review article, commentary article, editorial, conference paper or other peer-reviewed article.
— Study is a grey-literature report from a recognized stakeholder.
— All types of methodology (quantitative, qualitative, mixed, etc.) are eligible.

Based on these criteria, two reviewers (first and second authors of this paper) then separately assessed eligibility via screening of titles and abstracts. Where at least one reviewer perceived the study eligible, it was included (50% necessary percentage agreement). In total, 239 articles were judged relevant by at least one reviewer. Full texts were retrieved on 2 February 2021. All reasonable attempts were made to obtain full-text copies of selected articles (if not openly accessible, then first via institutional access privileges, and if that failed via inter-library loans or contacting the authors directly), whereupon a further 31 articles were removed as their full text was in a language other than English or the full text could not be obtained. Following this, 208 articles were carried forward to the next stage.

Full texts of the remaining articles were then examined by the first and second authors to determine to which research sub-questions the article was relevant. This literature was then delegated among authors according to topic.[3]

## 2.3. Charting the data and summarizing results

Each author responsible for that theme then appraised the full text to determine whether the study contained relevant evidence or discourse. Where it did ($n = 105$), a data charting form (table 1) was followed to electronically capture relevant information from each included study.

---

[3]T.R.-H.: general factors, open evaluation; S.R.: Open/FAIR Data, policy aspects; N.P.: Open Access; T.K.: Open Methods; N.L.C.: society aspects; A.F.: industry aspects.

**Table 1.** Data charting form.

| data chart heading | description |
|---|---|
| author | name of author(s) |
| date | date article sourced |
| title of study | title of the article or study |
| publication year | year that the article was published |
| publication type | journal, website, conference, etc. |
| DOI/URL | unique identifier |
| relevance to which study questions | Open Access, Open/FAIR Data, Open Methods, Open Evaluation, society, industry, policy |
| key findings, including study aims, details, design and data sources (where relevant) | noteworthy results of the study that contribute to the scoping review question(s). Where relevant, overview of the main objectives of the study. Type of study, empirical or review, etc. Notes on methods used in the study (whether qualitative or quantitative, which population demographics studied, etc.). Detail the data sources |

Authors then used further snow-balling and specific keyword search using web search engines and other sources to identify further relevant peer-reviewed material as well as grey literature, yielding a total of 163 items identified by other methods, in addition to the 105 items identified via Scopus, for inclusion. All results were then exported to a single library in the Zotero open source reference management software. Data charting was collated in a combined CSV file. The co-author responsible for each theme then drafted an initial narrative summary of the evidence. These sections were then compiled by the lead author and revised into a full first draft, which was then shared with all co-authors, who worked collaboratively to revise the study and fill any perceived gaps in evidence and argument.

These methods were pre-registered in advance on 22 December 2020 (https://osf.io/t6uy9/). All materials and data are available at doi:10.5281/zenodo.4936202. This resulting paper deviates slightly from the pre-registration in broadening the title and framing of the paper from a narrow focus on the Matthew effect to dynamics of cumulative advantage and threats to equity more broadly, in order to better reflect the scope of the pre-registered search queries and the resultant paper.

# 3. Results

The following sections present our synthesis of this literature. Since the many potential benefits of Open Science have been well-argued elsewhere [43–46], our presentation here necessarily focuses in greater depth on those areas where Open Science implementation potentially endangers the aim of greater equity in science. This emphasis should not be interpreted as signalling that the authors believe that the negatives outweigh the positives. The presentation of the results is in a descriptive format (narrative summary) to align with the study objectives and scope of the review, and phenomenon-oriented according to the various dimensions of Open Science: Open Access, Open Data, FAIR Data, Open Methods and Open Infrastructure, Open Evaluation, as well as Open Science's interfaces with society, industry and policy. It begins with some overarching issues which apply generally across the dimensions of Open Science.

## 3.1. Overarching issues concerning inequity in Open Science

Open Science is aimed in part to counteract inequity. Opening access to publications enables readership beyond those privileged by journal subscriptions [47,48]. Data sharing and open methods allow reuse beyond the narrow networks of existing collaboration [49]. Greater transparency in processes of evaluation might eliminate bias in selection procedures [50,51]. Participatory processes of Citizen Science could make scientific endeavours more inclusive and understandable for large audiences [18].

But, as Chin *et al.* [52] remind us, 'transitioning to open research involves significant financial costs'. Open Science relies upon local training and support, as well as infrastructure and resources. Even in well-resourced regions such as Europe [53,54] and the USA [55], readiness-levels of training and support infrastructure among nations and institutions are highly diverse. These disparities are, of course, even greater in what Siriwardhana [56] terms 'resource-poor' settings. Given that Open Science practices depend on underlying digital competences [57], the continuing realities of the digital divide [58] have real effects on participation in an Open Science world.

Implementation of Open Science must also be supported by policy. As Prainsack & Leonelli [59] argue, 'open science is a political project to an even greater extent than it is a technological one'. In Europe, the Open Science policy landscape is highly variable across nations, funding organizations and institutions [60]. Policy priorities shape incentive structures and resource allocation, and hence drive different implementation strategies. Open Science perhaps began as a grassroots movement of scholars, but its quick uptake into national and institutional policy has seen it linked to wider goals, including economic growth. In Europe, the European Commission (EC) has been a driver of Open Science [61]. But as an influential 2016 EC publication makes explicit, this interest is at least partly motivated by Open Science's perceived potential to maintain and promote Europe's 'competitive edge in global knowledge markets in the information age' [62]. To which we must naturally ask, at whose expense?

In the light of this, we should take seriously a strand of critique which links Open Science to broader trends to reshape the academy under neoliberal principles to emphasize market principles of competition, foregrounding its economic role in training the workforce and fostering new products and services, at the expense of the social mission to provide upward mobility for marginalized populations [63–66]. For some critics, Open Science has the potential to merely fuel these developments. In the words of Tyfield [67], Open Science's legacy may be defined by its 'effects on the construction of a new moral economy of knowledge production', meaning the marketization of science for the benefit of corporations [67]. Similarly, for Mirowski, Open Science will result in a 'platformization' of science—for-profit firms colonizing the research landscape with a host of tools, seeking to construct 'the One Platform to Rule Them All', and the research process being subject to increasing division of labour wherein smaller and smaller chunks are made objects of public scrutiny (e.g. open projects, open laboratory notebooks) [41]. Such developments could see, in the words of Kansa [68], the 'cause of "openness" subverted to further entrench damaging institutional structures and ideologies'.

## 3.2. Inequities in Open Access

The rationale for Open Access (henceforth OA, whereby scientific publications are distributed online, free of cost or other access barriers under open licensing conditions) to research publications is often centred around the democratization of knowledge—what Fitzpatrick [69] calls the 'ethical desire' to remedy an imbalance between information 'haves and have nots' (cf. [47,48]). OA is posited as boosting return on investment [70] and as a solution to inequity to information access in regions [71–76] and disciplines, especially to improve public participation in conversations related to social challenges like health, education and agriculture [77–82]. Yet, similar to Open Science more broadly, OA is also not a 'movement with a coherent ideological basis' [83]. Democratization is but one aim among others, including efficiency gains through speeding dissemination and potentially lowering publishing costs [47]. The diverse ethical, political and economic priorities motivating these aims, in turn, present a range of possible routes to OA implementation. A crucial issue in this regard has been the extent to which policies favour publishing in OA journals (Gold OA) over author self-archiving of non-OA publications in OA repositories (Green OA).

Gold OA can be supported via a multitude of business models, including consortial funding (also called 'Diamond OA'; cf. [84]) or volunteer labour [83], but many OA journals and publications are financed via article processing charges (APCs). The APC model is controversial since the benefit of OA (free readership) is offset by a new barrier to authorship at the other end of the publication pipeline. In this regard, the extent to which OA policy has been driven by richer, Global North nations risks reshaping scholarly communications to enable access but still foster exclusion. As the costs of APCs are usually borne by institutions or research funders (via project funding), those with fewer resources are disadvantaged [56,85]. The UK government's 2012 decision to clearly favour 'publication in open access or hybrid journals, funded by APCs, as the main vehicle for the publication of research' [86] can be seen as a watershed moment for APC Gold OA. More recently, the related funder-led initiatives OA2020 and 'Plan S'[4] were initially accused of ignoring experiences and

interests of developing nations and lacking support for 'the advancement of non-commercial open-access initiatives' [87]. Although Plan S has arguably somewhat corrected course here [88,89], the overall impact of Plan S remains to be seen.

APC-based OA hence risks stratifications of publishing as well-resourced researchers can cover even the highest APCs while less well-resourced researchers cannot [90–100]. Even in well-resourced areas like the UK, rising costs [101] of APCs is recognized as an issue which will mitigate OA's net benefits [102]. Although many publishers offer fee-waivers and discounts to authors unable to pay, restrictive terms and administrative burden, as well as the extent to which they seemingly place authors in the position of asking for 'charity', mean that they are often criticized as a weak response to an urgent issue [103,104]. This may have effects on specific demographics. For instance, Niles *et al.* [105] found that women tend to take cost into consideration significantly more than do men when deciding where to publish. This issue is made especially pressing by the citation advantages linked to OA publishing [46,106]. Usually seen as an important driver for motivating OA, in the light of equity such an advantage may in fact merely further fuel a classic Matthew effect, privileging those actors with the resources to pay for OA in the most prominent journals.

Given the problematized [107] but persistent association between journal prestige (often quantified via the journal impact factor) and publication quality, it is especially problematic, then, that publishers often charge more for high-impact journal publications [95,99].[5] In addition, APC rates are highly variable across disciplines [109] and regions [90], and are especially problematic in what Eve has referred to as the 'dry climate' of funding in the social sciences and humanities [110].

The APC model, combined with the pressure to 'publish or perish' [111], has also helped give rise to what is termed 'predatory' journals and publishers, who collect APCs for publishing articles with little or no editorial rigour [112]. The extent of the 'predatory publishing' problem has been argued as overhyped by traditional publishers eager to discredit OA [113]. Indeed, according to Shen & Björk [114], it is 'highly contained' to a few countries. Yet even if limited, there is a problem, nonetheless. Authors from developing nations or with less-developed competences in English, already known to be disadvantaged in traditional publishing [93,115–118] as well as early career researchers with limited publishing experience, are known to be especially affected by predatory publishing [119–124]. Given the stigma attached to publishing in these venues, predatory publishing, therefore, poses a risk to the development of early career and developing-world researchers and potentially contributing to what Collyer terms 'two separate publishing circuits, leading knowledge produced in the Global South to be "systematically marginalized, dismissed, under-valued or simply not made accessible to other researchers"' [125].

Such stratifications in publishing, favouring traditionally advantaged actors (including for-profit publishers), will only exacerbate historical inequalities [126] and undermine wider aims of Open Science. Hence, as Nyamnjoh argues, for 'open access to be meaningful … questions of content and the epistemological, conceptual, methodological and contextual specificities that determine or impinge upon it are crucial' [127]. We, therefore, agree with Czerniewicz [128] who argues that such consequences are the result of too narrow a focus on achieving OA *per se*, by whichever means, without acknowledging 'the inequitable global power dynamics of global knowledge production and exchange'. Rather, she suggests, we must broaden our focus 'from access to knowledge to full participation in knowledge creation and in scholarly communication'.

## 3.3. Inequities in Open Data and FAIR Data

Data sharing has been linked to increased citation rates [129], economic growth [46], transparency [130], reproducibility [131], improved research quality [132], reuse [133] and efficiency [134]. These benefits have often been dissociated from specific research contexts, however. Hence, some suggest that the Open Data movement has overestimated the homogeneity of research environments [135], resulting in a generalized 'assumption that all scientists will benefit from releasing data, no matter where they are based' [136]. Such an assumption fails to appreciate that conditions for making data available differ across disciplines [137] and regions [136]. The concept of data as decontextualized facts removable from the context that underpins the idea of data sharing has come under criticism of late as scholars are beginning to appreciate that data are situated and mutable [138]. In fact, it has been suggested [139] that 'data' is an umbrella term whose meaning changes with context: with the specificity of the research purpose, with the scope of data collection and with the goal of the research [139].

---

[5]*Nature* even recently agreed terms to charge equivalent APCs of up to €9500, for example [108].

Data sharing occurs for many reasons, including reproducibility of published research, enabling others to ask new questions and making publicly funded research publicly available [134,139]. The situated nature of research practices means motivations vary across research contexts [140]. Hence data sharing is viewed more favourably in some fields than others [138]. To date, much work on data practices has arguably been led by the (biomedical) sciences [141], with less attention to other disciplines. For example, the FAIR principles (to make data Findable, Accessible, Interoperable and Reusable, but importantly not open) [49] were heavily inspired by life sciences research. This initial focus has carried over to the scope of empirical work on their adoption [142]. But contexts differ depending on issues like the readiness-level of data formats [143] and whether the research involves human subjects [144–146]. For these disciplinary reasons, a blanket appreciation of Open Data as inherently democratic is problematic [147,148]. One-size-fits-all policies may, therefore, disadvantage those disciplines and actors less able to participate, and further add to the prioritization of STEM subjects.

Data inequalities are also cumulative, shaped by individual and community characteristics, access to infrastructure, and political and economic factors [149], affecting the abilities of different groups to partake in the 'gift economy' [150] of academia. As ethnographies of (non-Western) research [142] show, access is not enough to guarantee that Open Data can be reused effectively because reuse requires not only access, but other resources such as skills, money and computing power [148]. Those working in environments where these are in short supply might be put at a disadvantage [136,142,151]. Additionally, making use of Open Data is closely linked to data literacy, potentially marginalizing those that cannot engage with data effectively [152–154]. Edelenbos et al. [155] argue that Open Data 'are particularly accessible to research institutes with more budget'. Hence, increasing evidence suggests that instead of levelling the playing field, Open Data might simply empower those already advantaged [149,156,157]. In this way, existing inequalities moderate the positive effects of Open Data which means they might be just a further mechanism whereby the rich get richer instead of leading to the democratization of knowledge.

In addition, the effects of data-intensive research on careers should be monitored for their outcomes regarding equity. Studies of authorship contributions to publications have found a clear gender divide [158]. Women are more likely to contribute to the investigation, data curation or writing of the original draft, whereas men are more likely to contribute to tasks associated with seniority (supervision, funding acquisition, resources). This division of labour and capital among researchers might reinforce existing hierarchies and cumulative advantages, in that additional workload involved with Open Data is frequently proportionally carried more by women.

These barriers to participation in Open Data are made ever more pressing by the citation advantages linked to data sharing [129]. Piwowar et al.'s results approximate the original conception of the Matthew effect through establishing a clear (not necessarily causal) connection between Open Data practices and citation advantage. Giving due attention to the various contexts within which data sharing does or does not happen is, therefore, paramount for meeting the goals of inclusivity and openness espoused by the Open Science agenda.

## 3.4. Inequities in open methods and open infrastructures

Open Science and in particular open methods, which involve practices like sharing analysis code, laboratory notebooks or preregistering analyses, hold the promise to counter current concerns regarding integrity in reporting and the reproducibility of research [44,159,160]. There are a few potential impacts on equity, however. First, it seems plausible to expect that better-resourced academics could be early adopters in terms of open methods [92]. Well-resourced institutions can provide the necessary setting to successfully integrate open practices into the research workflow more easily. Well-resourced and high-status actors tend to be early adopters in general [161] and methods such as pre-registration or sharing of research notes and code need additional training, effort and access to infrastructure to be implemented correctly. To the extent that transparency in research is increasingly becoming a benchmark for quality [162], these well-resourced players will potentially have an advantage.

Secondly, the meanings and limits of openness are not uniform across disciplines. Calls to increase the reproducibility of research findings originated in specific fields, most prominently biomedicine and psychology [163,164]. While normative calls for increased standards in reporting are diffusing to further disciplines, it must be recognized that the notion of reproducibility is not equally applicable everywhere. Methodological approaches as found in mathematics, information sciences and computer

sciences are clearly better suited for reproduction, based on their reliance on statistics and their level of control over the research environment. On the other hand, qualitative approaches as found in parts of the humanities, history and social sciences are more difficult to assess in terms of reproducibility [162,165]. In the same way that the FAIR data principles have been designed primarily for quantitative data (data that does not rely on human subjects), extending the standards of quantitative methodologies to qualitative approaches in attempts to make them more 'scientific' may 'obscure unavoidable interpretive work' [166] or further devalue qualitative approaches which cannot meet such criteria. This could then also reproduce existing inequalities of race and gender, since quantitative-focused fields, in particular STEM, are known to favour white men, and academic participation among women and racial and ethnic minorities is higher within the humanities and social sciences [167].

Finally, open methods are heavily infrastructure-dependent, reliant on networked online platforms and other e-Infrastructures. Concerns have been raised about the extent to which privately owned platforms may frustrate the aims of Open Science [41,168,169]. Recent years have seen major publishing corporations like Elsevier, Wiley and Springer (via subsidiary Digital Science) rush to capture researcher workflows through a host of proprietary tools which often eschew interoperability in favour of intraoperability with their own product suites [168,170]. Hence, there is a growing recognition of the need for Open Science infrastructures to themselves be open source and community-governed to ensure they (and the data they generate) remain community resources responsive to community needs [168,171–174]. As Hall has noted, the Open movement is 'in danger of being outflanked, if not rendered irrelevant, as a result of our media environment changing from being content-driven to being increasingly data-driven. For the data-driven world is one in which the data centre dominates' [175].

Yet sustainability and governance models for open infrastructures remain unclear. Funding often comes from competitive project grants whose bureaucratic funding logic often requires inflexible and shorter-term project work-plans be satisfied at the expense of longer-term planning, agile development and broader interoperability within the infrastructure ecosystem [176]. In addition, open infrastructures often rely on voluntary contributions from open source communities [177]. In this regard, we must first ask: who is building and for whom? According to Ehls [178], open-source communities generally skew heavily young (average age 27–32) and male (91–98%). How this homogeneity of contributors may influence issues like gender-bias [179] in design of open source tools should be monitored. In addition, that mainly younger people may be contributing requires us to examine the ways in which contributions are rewarded. In the prestige economy of academia, open source contributions are heavily undervalued in promotion, review and tenure procedures, where publications still dominate [180]. Hence, we might say that open infrastructures are very often reliant upon the unpaid contributions of early career researchers, whose precarious employment conditions [181] mean that their time could be better invested in terms of career advancement [182]. Appropriate credit and recognition (e.g. during evaluations for hiring, promotion and review) of work involving open methods and open infrastructures are, therefore, key for attaining equitable outcomes in the uptake of open methods and development of infrastructures.

## 3.5. Inequities in open evaluation

Open evaluation identifies the ways in which Open Science principles of transparency and inclusivity can be applied to the evaluation of research and researchers via peer review or metrics.

Peer review, assessment of research outputs by external experts, is the gold standard for evaluation and selection in scholarly publishing, conferences and funding allocation, but is often criticized as inefficient, unreliable and subject to bias [183]. Open peer review applies Open Science to reform peer review in various ways, most prominently by removing reviewer anonymity or publishing review reports [51]. A major supposed advantage is increased review quality. Yet opponents counter that this may compromise review processes, especially considering power imbalances, either by discouraging full and forthright opinion or opening especially early career reviewers to potential future reprisals from aggrieved authors later on [184]. Given that a recent study found that publishing reports does not compromise review quality, at least when allowing anonymity [185], it seems the issue of de-anonymizing reviewers is the main issue. By contrast to other elements of open peer review, opening reviewer identities is not favoured by researchers [186].

Research metrics are used throughout research and researcher evaluation processes, usually based heavily on counting citations, often aggregated via mechanisms like the h-index or Journal Impact Factor. However, citations have been widely criticized for being too narrow a measure of research

quality [50,187,188]. The application of particularistic standards is especially perilous for early career researchers who have yet to build their profile. By using citation metrics to evaluate research contributions, the Matthew effect leads to the self-reinforcement of initial positive feedback [24]. Moreover, indicators such as the impact factor are highly reactive [189] and therefore exacerbate a quasi-monopolization of resources (prestige, recognition, money) in the hands of relatively few institutions and individual researchers.

The rise of the 'social' web in the mid-2000s soon gave rise to calls for 'alternative metrics' or 'altmetrics' to be part of balanced research assessment by aggregating additional online impact measures such as tweets, likes, shares, bookmarks, blogs and press coverage [190,191]. With their intention to expand the scope of research assessment using new sources of web data, altmetrics have been associated with the move to Open Science [192,193]. Perceived advantages include the speed of data availability and ability to track outputs beyond publications [190,191].

Given general agreement on the limitations of, and over-reliance on, citations, broadening the range of possible data sources for research evaluation is welcome. Yet altmetrics themselves have been criticized for a lack of robustness and susceptibility to 'gaming'; disparities of social media use between disciplines and geographical regions; reliance on commercial entities for the underlying data; indicating 'buzz' or controversy rather than quality; underrepresentation of data from languages outside English; and underrepresentation of older papers [183,192,194–199]. Each of these could have consequences for altmetrics' efficacy as tools of equitable research assessment and advantage some at the expense of others. In addition, the very idea of expanding the number of metrics used for the assessment can be critiqued as merely perpetuating the 'tyranny of metrics'. Ryan [200], for example, argues that metrics *per se* are tools of surveillance, enclosing academic freedoms by furthering a neoliberal idea of the academic as a competitor in a game of visibility, and promulgating a negative culture of competition that is the root of much ill in the academy (cf. [41,67]).

## 3.6. Inequities in Citizen Science

Strengthening the relationship between science and society is a key pillar of Open Science. Citizen Science seeks to foster inclusion in knowledge production by involving the public in scientific processes. Practices range from research where the public contribute data to scientific projects via crowdsourcing platforms, to 'extreme citizen science' or 'strongly participatory science', wherein members of the public participate in all aspects of research and are able to make valuable use of the research results in ways that benefits their lives and communities [201,202]. It is the latter, rooted in traditions of participatory research, that most directly and strongly challenges the dynamics of inequality within academia [201,203]. Participatory research is directed by a self-reflexive, critical, ethics-focused approach to research design, conduct, distribution of labour [204] and financial benefits [204–206], outputs and impacts [205–209]. Questions of equality, justice and equity are often centred [202,206,208,210–212] and the knowledge produced reflects the lived experience and situated expertise of the participants and communities upon which the research is focused [202,203,213,214]. Further, those that participate in Citizen Science and participatory projects develop scientific expertise that would otherwise be unavailable to them, making them more effective democratic actors who are able to challenge policies, civic expertise and corporate power in pursuit of justice and equality [202,208,211,215–219].

Nevertheless, Citizen Science may also in some circumstances perpetuate inequalities. When participants serve merely as free data collectors, those who primarily benefit are researchers [209,220–222]. Where such practices bridge the Global North and South, data extraction absent anything else echoes colonial exploitation [205]. Additionally, there are issues of biased inclusion in terms of the populations that are invited to participate in traditional Citizen Science [223,224], with the most marginalized groups likely to be left out [220,221,225,226]. Likewise, there is biased participation in the crowdsourcing of information [227]. Finally, approaches taken by well-intentioned researchers may also reinforce existing inequalities, like when a paternalistic stance is taken towards participants (see the use of 'provide voice' in Hendricks *et al*. [211]); research seeks to prove the quality and validity of citizen-generated data, thereby reifying the expertise divide between scientists and the public [210,220,221]; or when science education is framed as a unidirectional resource coming strictly from scientists (e.g. [228]). All these have implications for equity in Citizen Science relating back to our themes of how cumulative advantage can impact participation in Open Science, especially related to issues of digital divide, differential levels of resources and skills and data equity. Again, it is non-dominant actors who are at risk. Shelley-Egan *et al*. [229] criticize such instumentalization of participants as demonstrating that in

Open Science 'publics operate as citizen scientists collecting or systematizing data without necessarily reflecting on or critiquing the broader institutional and societal frameworks for uptake'. In summary, these findings point towards a biased inclusion of populations invited to participate in Citizen Science projects, which tends to perpetuate the divide between experts and publics and raises questions of representation and equity: Whose voices are represented in Citizen Science projects, and for which reasons?

## 3.7. Inequities at the interfaces of Open Science and society, industry and policy

The societal impact of research has been an increasing factor in policymaking throughout academia in recent years, especially in research evaluation exercises. The drive for Open Science and new forms of knowledge production has been intimately linked, at least at the policy level, with this agenda [40]. For instance, the EU believes Open Science 'instrumental in making science more responsive to societal and economic expectations' [62]. However, we can question the terms on which such responsiveness has thus far been sought. As we shall see, there are crucial asymmetries which potentially compromise, or at least restrict, Open Science's potential to fully realize equity in research.

Firstly, we must point out that achieving impact is a resourceful activity. A dominant theme in our study so far is that enabling access is not enough and this is also the case for societal impact. In policy uptake of scientific knowledge, for example, it has been suggested that Open Science will help by making scientific resources more readily available to policy-makers and other policy actors [46,230,231]. However, such uncritical narratives of openness fail to address structural barriers in knowledge production. Firstly, more high-profile OA output from established actors may lead to further over-representation of knowledge produced by dominant groups [140,232,233]. Perhaps more importantly, though, in addition to access, relationships between academics and policy-makers are a main factor in policy uptake of science [234]. Policy-makers, with limited time to make decisions and to seek advice, heuristically rely on (personal) networks of experts that have previously contributed to policymaking [235]. Researchers' and intermediaries' translation skills (elevator test) are particularly important in this regard [236], with tailored messages a key driver of uptake [237]. Building relationships and fine-tuning messaging require time and effort, and can be significantly bolstered by support structures within research-performing institutions [238]. Hence, researchers with access to such support are advantaged in ensuring uptake.

Knowledge-transfer services are also vital in fostering the uptake of scientific resources in the industry. Here, indicative evidence shows that Open Science might have a positive economic impact. A recent synthesis [45] summarizes the literature to find that Open Science can help industry-uptake through (i) efficiency gains through easier access to publications [239–242] and data [243–245], as well as reduction of transaction costs via collaborative approaches [246,247] and lower labour costs or increasing productivity [240,248–250], and (ii) enabling new products, services or collaborative possibilities [251–253]. However, evidence points towards firms (particularly small and medium-sized enterprises) lacking necessary skills such as information literacy to fully benefit from open resources [240,254,255]. Given the above discussion of the underlying digital competences necessary for uptake of FAIR Data, this will be especially acute for those on the wrong side of the digital divide. Again, the most well-resourced stand to benefit most. Ironically, this fact extends even to the demonstration of impact. As Bornmann [183] points out, institutional societal impact is often assessed based on case studies which are 'expensive and time-consuming to prepare'. This all suggests that uptake of scientific knowledge, even in an age of Open Science, remains prone to dynamics of cumulative advantage.

There are more fundamental dimensions of asymmetry. Regarding industry, Fernández Pinto [15] argues that current Open Science policies risk perpetuating the commercialization of science in three ways. Firstly, the focus on opening *publicly* funded research allows industry the 'privileged position of adopting openness as they see fit', adopting openness where it is commercially attractive or improves public image (cf. [256]) and ignoring it in less favourable circumstances, such as where findings may impact sales. Secondly, the policy focus on Open Science's potential to spur innovation means that the science–industry connection is deepened without critical reflection on the 'epistemic and social justice challenges' of private sphere research, including scandals in corporate-sponsored scientific research (cf. [257–259]), conflicts of interest and their influence on research work (cf. [260]) and the ways in which strong intellectual property regimes might inhibit or corrupt scientific research (cf. [261]). Finally, repeating the point made above, the networked and platform-dependent nature of Open Science enables commercial interests to increasingly control and commodify research processes [41] (cf. [67]). For Fernández Pinto, therefore, the issue lies in the unequal terms on which Open Science engages

industry, with the latter privileged. This asymmetry in favour of industry can in turn can be seen as an endorsement within Open Science of the marketization of science and the specific neoliberal vision of the academy which underlies it (cf. [214]).

Shelley-Egan *et al.* [229] identify another asymmetry, this time at the expense of the public. The authors compare the Open Science agenda with that of the seemingly complementary responsible research and innovation (RRI). RRI is a science policy movement, especially prominent in Europe, which seeks to better integrate science with society [262,263]. Deriving from a tradition of participatory research, technology assessment and anticipatory governance, RRI aims to avoid 'expert hubris' and unintended consequences. Shelley-Egan *et al.* argue that in contrast to RRI, Open Science's ambitions are more pragmatically focused in terms of engagement with the public. Whereas 'RRI's approach to opening up extends to an invitation to publics to co-define the aims and means of technical processes in order to increase their alignment with public values', 'Open Science restricts ambitions for opening up to adjustments and improvements to processes based on quality criteria ultimately rooted in the existing research system'. Open Science is thus seen as insufficiently critical of the value and direction of science. It is also seen as failing to fully appreciate 'societal voices and citizens as legitimate conversation partners and beneficiaries of technology and knowledge', engaging publics on asymmetrical terms that seek mere dialogue between 'technical experts and societal voices'. Hence, 'RRI focuses more on producing (ethically and societally) "good" outcomes than on resulting in the (epistemically and functionally) "best" outcomes, while OS for its part remains agnostic about the former and concerns itself almost entirely with the latter, and more often concerns itself with issues of efficiency, optimization, integration and potential'. The authors suggest that this 'pragmatism and instrumentality' of Open Science leaves it in line with prevailing political and institutional (i.e. neoliberal) aims.

Such a purely technocratic definition is no doubt at odds with the view of Open Science held by many advocates. Yet the equivocal nature of 'Open Science' means that such readings are at least plausible, and hence should be taken as a call for the Open Science community to more fundamentally appraise the way its priorities are presented, and the deeper ways in which societal voices can be engaged as equals in setting research agendas. It further illuminates the ways in which the radicality of Open Science can be questioned, and the extent to which Open Science merely enables the further neoliberalization and commodification of research knowledge.

# 4. Discussion

This synthesis of evidence is intended to focus attention on the ways prevailing capacities, resources and network centralities—combined with structural inequalities and biases—can help shape Open Science outcomes. Inequalities and dynamics of cumulative advantage pervade modern scholarship, and our results show that despite its potential to improve equity in many areas, Open Science is not exempt. Merton advises that cumulative advantage directs our attention to 'the ways in which initial comparative advantages of trained capacity, structural location and available resources make for successive increments of advantage such that the gaps between the haves and the have-nots in science (as in other domains of social life) widen until dampened by countervailing processes' [23]. From the above synthesis we can observe that this mechanism is at work at various levels throughout Open Science, potentially endangering equity. We have identified key areas for concern, summarized in table 2.

These issues can, in turn, be attributed in various ways to some fundamental concerns:

— *Ambiguity and politics*: Open Science is an ambiguous and deeply political concept. We should not expect that all of the diverse practices, much less their many possible routes to implementation, that fall under this umbrella term should accord in every aspect. Equity is one aim among others and may conflict with others like efficiency and transparency. The policy-driven focus on Open Science's potential to fuel economic growth, in particular, seems designed to maintain the economic advantage and hence conflicts with wider aims of global equity. Moreover, narrow focus on specific elements of Open Science, at the expense of a more holistic view of the (dys)functioning of the scientific system as a whole, may exacerbate such factors.
— *Resource-intensity and network effects*: Cumulative advantage relates to logics of accumulation and preferential attachment based on network positionality and possession of resources. The resource-intensive and networked nature of Open Science means it is also vulnerable to these logics. Explicitly linking authorship channels to possession of resources potentially stratifies Open Access publishing. The expensive infrastructures and training necessary to participate in engaging with Open Data and methods means those privileged in these regards are primed to benefit most, at

**Table 2.** Summary of identified areas of concern for equity in Open Science.

| aspect of Open Science | area for concern | group(s) most affected |
|---|---|---|
| general factors | costs of participation: Open Science is resource-intensive in terms of infrastructure, support, training | less well-resourced institutions and regions |
| | political agendas: Open Science requires political will, but political agendas shape Open Science implementation. Especially where economic growth is a stated ambition, this may be problematic | regions and institutions without such political backing, or where political goals promote inequitable Open Science implementations |
| | neoliberal logics: Open Science seen as potentially entrenching structures and ideologies of neoliberal commodification and marketization of research knowledge as an economic resource to be exploited rather than as a common good for the well-being of humanity | science *per se*, but especially those disciplines and researchers that do not fit this agenda |
| Open Access | discriminatory business model: APC-based OA is exclusionary and risks stratifying authorship patterns | less well-resourced researchers, institutions and regions. May also impact specific demographics, including women |
| | predatory publishing: limited issue which nonetheless primarily adversely affects non-dominant groups | authors from developing nations and early career researchers |
| Open Data and FAIR Data | situatedness of data practices: data practices are highly context-dependent, meaning one-size-fits-all policies risk privileging some disciplines | qualitative researchers and disciplines |
| | cumulative nature of data inequalities: creating and exploiting Open Data is strongly linked to access to infrastructure and data literacy | less well-resourced researchers, institutions and regions |
| | citation advantages of Open Data: Open Data seems linked to increased citations and hence early adopters benefit (Matthew effect) | less well-resourced researchers, institutions and regions |

(*Continued.*)

**Table 2.** (Continued.)

| aspect of Open Science | area for concern | group(s) most affected |
|---|---|---|
| Open Methods and Open Infrastructure | transparency as a benchmark for quality: open methods require additional training, effort, infrastructure. Well-resourced and high-status actors may potentially have an advantage | less well-resourced researchers, institutions and regions |
| | reproducibility as a *sine qua non* for research: relatedly, meanings and limits of openness not uniform across disciplines. Uncritically extending quantitative standards methodologies may obscure necessary interpretive work or further devalue qualitative approaches | qualitative researchers and disciplines |
| | platform-logic of Open Science: reliance on privately owned platforms may frustrate the aims of Open Science and increase surveillance capitalism in academia | science as a whole |
| | lack of reward structures for contributions to open infrastructure: Open Science seems at risk if it relies on closed and proprietary systems; yet open infrastructures often rely on short-term project funding or volunteer labour which is not properly rewarded within current incentive structures | early career researchers |
| Open Evaluation | open identities peer review: peer review where reviewers are de-anonymized may either by discourage full and forthright opinion or opening especially early career reviewers to potential future reprisals from aggrieved authors later on | erly career researchers, others from non-dominant groups |
| | suitability of altmetrics as a tool for measuring impact: altmetrics criticized for: lack of robustness and susceptibility to 'gaming'; disparities of social media use between disciplines and geographical regions; reliance on commercial entities for underlying data; indicating 'buzz' rather than quality; underrepresentation of data from languages outside English; exacerbating tyranny of metrics | all, especially non-English language research and areas where social media use is less pronounced |
| Citizen Science | logics of participation in Citizen Science: evidence of biased inclusion in populations invited to participate; potential for data extraction absent anything else to echo colonial exploitation | the public, especially marginalized groups |
| interfaces with society, industry, policy | resource-intensive nature of translational work: making outputs open is not enough to ensue uptake and societal impact. The importance of (resource-intensive) translational work means richer institutions and regions may still dominate policy conversations | less well-resourced researchers, institutions and regions |
| | privileging of economic aims: the terms on which Open Science engages industry is asymmetrical, perhaps reflecting the importance of economic growth as a key aim. Industry is free to participate (or not) in open practices, as it suits them | science as a whole, but especially those domains not easily exploited by commerce |
| | exclusion of societal voices: Open Science's terms of inclusion of publics is accused of 'instrumentalism' and asymmetry (experts/public) | the public |

least initially. The importance of such underlying competences means that ensuring access is not enough to ensure equity of opportunity in an Open Science world absent broader measures to overcome the digital divide. In addition, the ways in which these underlying competences interplay with actor attributes to shape logics of participation in networked communities means that there are concerns for who is privileged by proposals to reform areas like peer review and research evaluation, as well as who contributes to open source tools and services.

— *Narrow epistemologies*: The term Open Science itself is often seen as exclusionary of the arts and humanities in the anglophone community. More pernicious, though, is the potential devaluation of epistemic diversity that attends Open Science's focus on transparency and reproducibility. Within the Open Science paradigm, the latter concepts are becoming almost synonyms for research quality itself, rather than just important means of assuring quality *in some domains*. Qualitative methodologies for which transparency is less possible and reproducibility less relevant may be further marginalized if this trend continues. If so, Open Science will only add to the further cumulative advantage of STEM subjects within the academy at the expense of social sciences and humanities.

— *Neoliberal logics*: In addition, certain central assumptions of Open Science seem to further promote the commodification and marketization of research. Making science more responsive to the market risks further intensifying competition at the expense of communalism. Further, platform-logic pervades Open Science's enabling infrastructures. Not only does this risk lock-in by commercial vendors, but logics of data accumulation and tracking further enframe researchers as something to be measured in a regime of surveillance capitalism. If so, Open Science may act to only further advantage the neoliberalization of academia, which is often identified as a root cause of many of the issues Open Science is claimed to fix.

## 5. Conclusion

The synthesis itself is subject to some key limitations, including some ironically linked to issues discussed. The authors all work in the Global North at relatively well-resourced institutions and are funded by an EC research grant whose conditions of application reflect a focus on the situation in Europe. Article search was primarily via databases (Web of Science and Scopus) which employ strict inclusionary criteria, and although this was combined with snow-balling and secondary literature searching, it still might be the case that our review has not captured all available evidence. Further, the language skills of the authors meant a pragmatic decision to only include articles in English. In addition, reviews are necessarily retrospective. Although we have tried to be as balanced as possible, these particularities of standpoint, resources and inclusion criteria no doubt influence our critique.

Our work here has been to scope the (English language) literature to date concerning threats to equity within the transition to Open Science. This is of course preparatory work and by no means the end of the story. Directions for future work may include first the extension of this study to cover literature in languages beyond English. An additional study using the same methodology but involving a multi-lingual team covering the major world languages could be envisioned, and the present authors would be happy to collaborate in such an endeavour. Secondly, the issues raised in this work deserve much more scrutiny and so future primary research work involving qualitative and quantitative approaches on these issues is desired. Finally, this work has aimed primarily to scope the issues involved and is not strongly normative in the sense of producing specific recommendations on what policy actions may be suggested to correct potential negative effects on equity in the transition to Open Science. Such recommendations are no doubt required.

We hope that the wider Open Science community will take these criticisms in the constructive spirit in which they are meant, as a springboard to help recognize and further address such issues. As stated earlier, none of this is meant to diminish the aims of Open Science *per se*, or negate the good that Open Science brings and has the potential to bring. Rather, it is to align ourselves with Fernández Pinto [15], who questions 'the particular way the ideal has been conceived and implemented by the Open Science movement, as well as the way it has been brought about through Open Science policies. In this sense, the faulty logic of open science that I aim to highlight … refers precisely to the inconsistency between the ideal and its current implementation'. Given its commonly held aim of increasing equity, any potential for Open Science to actually drive inequalities must be taken seriously by the scientific community in order to realize the aim of making science truly open, collaborative and meritocratic.

Data accessibility. Protocol: https://osf.io/t6uy9/. Data: https://doi.org/10.5281/zenodo.4936203.

The data are provided in the electronic supplementary material [264].

Authors' contributions. Conceptualization: T.R.-H., S.R.; data curation: T.R.-H., S.R.; funding acquisition: T.R.-H.; investigation: T.R.-H., S.R., N.L.C., A.F., T.K., N.P.; methodology: T.R.-H., S.R.; project administration: T.R.-H., S.R.; resources: T.R.-H.; supervision: T.R.-H.; validation: T.R.-H.; writing—original draft: T.R.-H., S.R., N.L.C., A.F., T.K., N.P.; writing—review and editing: T.R.-H., S.R., N.L.C., A.F., T.K., N.P.

All authors gave final approval for publication and agreed to be held accountable for the work performed therein.

Competing interests. We declare we have no competing interests.

Funding. This work was supported by the project ON-MERRIT, funded by the European Commission under the Horizon 2020 programme (grant no. 824612). The Know-Center is funded within COMET—Competence Centers for Excellent Technologies—under the auspices of the Austrian Federal Ministry of Transport, Innovation and Technology, the Austrian Federal Ministry of Economy, Family and Youth and by the State of Styria. COMET is managed by the Austrian Research Promotion Agency FFG.

Acknowledgements. The authors gratefully thank the following for valuable comments on earlier drafts of this article: three reviewers for *Royal Society Open Science*, Bernhard Wieser (via email), Benedikt Fecher (via email), Daniel Lakens (via Hypothesis), Egon Willighagen (via Hypothesis) and Cameron Neylon (via Twitter). Any errors remain entirely the authors' own.

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
