## [Peer Review File · Royal Society Open Science]

Review History

RSOS-211032.R0 (Original submission)

Review form: Reviewer 1

Is the manuscript scientifically sound in its present form?

No

Are the interpretations and conclusions justified by the results?

No

Is the language acceptable?

Yes

Do you have any ethical concerns with this paper?

No

Have you any concerns about statistical analyses in this paper?

No

Recommendation?

Major revision is needed (please make suggestions in comments)

Comments to the Author(s)

In "Dynamics of Cumulative Advantage and Threats to Equity in Open Science: A Scoping Review" the authors present a set of criticisms or points that have the potential to exacerbate inequalities in the scholarly community. Some of them are very valid and will be important for readership to be aware of. Some of the criticisms are difficult for me to comprehend. The authors wisely do not attempt to weigh the relative costs and benefits of OS. However, they take that mentality a bit too far in my opinion, and the reader is left to wonder what to do with the information provided. Is awareness of the potential problem sufficient? In some cases yes, but in most cases there should be some recommendation to revise how knowledge generation is valued, or how policies should be updated to provide broader guidance for researchers in various regions or disciplines.

I believe that I am part of the target audience of this paper, as I work closely with policy makers on open science initiatives (as a matter of personal policy, I disclose my name and role in the provided fields). Acknowledging my perception, my goal with this review is not to serve as a gatekeeper, as I think that criticisms should be widely shared. But I mostly want to help the authors reach me in order to provide a persuasive set of recommendations or lessons to be applied. That may result in a rather reductive view of the utility or purpose of this scoping review, but it nonetheless is what I would need to see in order to have it affect my opinion, which I think it will be capable of doing.

Specific questions or concerns are raised below, after quotes from the text.

The authors state: "In light of this, we should take seriously a strand of critique which links Open Science to broader trends to reshape the academy under neoliberal principles to emphasise market principles of competition, foregrounding its economic role in training the workforce and fostering new products and services, at the expense of the social mission to provide upward mobility for marginalised populations"

I don't follow how OS exacerbates this trend. The trend itself is clear, and it is clear how those forces could use OS, but it is not clear whether there is any feature of OS that makes it more or less susceptible to the trend. I know the authors are not trying to give a weighing of the costs and benefits of OS, but a contrast in this section (between factors that are more susceptible to these pressures under an OS regime than a "traditional" regime) would clarify to the readers (both those who are skeptical of the authors claims and those who are merely trying to follow their logical) how the authors are pointing out the particular risks associated with OS.

The Open Access section does not appear to add anything particularly new. It does synthesize the arguments, but the tension between OA and the cost that transfers to knowledge creators from knowledge consumers is fairly well known. As with my previous acknowledgement that the authors are not (and should not) attempt to weigh the relative costs and benefits, there is nonetheless a missing opportunity to point to the for-profit publishing organizations as the primary forces that are exacerbating the problem, and that no matter who pays (authors or readers), the winners are the publishers and the losers are those who are least able to afford to publish or read. OS in general seems to have little overall effect, just a shift that publishers are responding to.

Open data. "...access is not enough to guarantee the utility of open data because the latter requires resources such as skills, money, and computing power [148]. Those working in environments where these are in short supply can fear that opening data sets will put them at a

disadvantage" I don't follow the logic of this argument, which seems to be one of the two major points of section 4.3 (the other being that open data is primarily modeled after quantitative, STEM research that does not rely on human subjects). It is true that access to data is not sufficient to recognize its utility, but it at least seems necessary. If skills, money, and computing power are in low supply, then opaque data or open data will both result in lack of some result. This almost implies that, since not everyone can use some data, then no one should be able to. Perhaps it does increase inequality by virtue of the fact two groups not conducting research are equal, compared to two groups where one conducts research and the other doesn't are not equal. But if that is the conclusion, then I would recommend that the authors be explicit about it and then discuss its implications. A recommendation from that could be that "equality" should not be used as a primary argument for OS, as it could motivate an unintended race to the bottom.

Section 4.4 "...first movers will potentially have an advantage." I think an important point would be to articulate how first-movers and the existing overly privileged knowledge producers overlap. I can see some readers would assume that "first movers" is synonymous with "privileged," however, I think many who identify as first movers would also identify as smaller players who are less privileged than their peers.

"or further devalue qualitative approaches which cannot meet such criteria." I think this is an important point that the authors should emphasize more. I agree that OS is created with a particular sub-set of methodologies and values them over others.

"The term Open Science itself is often seen as exclusionary of the arts and humanities in the anglophone community. More pernicious, though, is the potential devaluation of epistemic diversity that attends Open Science's focus on transparency and reproducibility." In order to address this concern, the OS community could either a) be more inclusive of the arts and humanities as knowledge producers or b) assert that many of the recommendations are designed specifically to improve how quantitative, experimental inferences are made. If "a" is pursued, it seems to make "b" more difficult, because one cannot apply the same methods or expectations to both (as the authors rightly point out).

Review form: Reviewer 2

Is the manuscript scientifically sound in its present form?

Yes

Are the interpretations and conclusions justified by the results?

Yes

Is the language acceptable?

Yes

Do you have any ethical concerns with this paper?

No

Have you any concerns about statistical analyses in this paper?

No

Recommendation?

Accept as is

Comments to the Author(s)

The researchers conducted a search of published and grey literature on the research area from January 2000 to the present, published in English, which yielded 3170 total results. Their research looking for kind of evidence and discourse which exist in the literature about the ways in which dynamics and structures of inequality could persist or be exacerbated in the transition to Open Science, across disciplines, regions and demographics, may contribute a lot ensuring in finding inequities in certain publications.

The researchers presented their goal in a clear way, used an adequate methodology, and composed the discussion and conclusions in a thorough way. In addition, they referred to the limitations of the study, following by a recommendation that “any potential for Open Science to actually drive inequalities must be taken seriously by the scientific community in order to realize the aim of making science truly open, collaborative and meritocratic.”

Based on the above, I recommend accepting the manuscript.

Review form: Reviewer 3

Is the manuscript scientifically sound in its present form?

Yes

Are the interpretations and conclusions justified by the results?

Yes

Is the language acceptable?

Yes

Do you have any ethical concerns with this paper?

No

Have you any concerns about statistical analyses in this paper?

No

Recommendation?

Accept as is

Comments to the Author(s)

The research topic is crucial for the scientific community.

The article is well written and highly informative. The information is presented in a balanced manner, which leaves full space for the reader to come to his/her independent conclusions. At the same time, the way in which it is presented stimulates reflections and thoughts, what is the naturally-expected outcome of this type of research.

I am convinced that this article needs to be made available to the scientific community as soon as possible.

Given that I do not find any faults requiring specific addressing, I strongly recommend that the article is published as it is.

I have two minor suggestions:

** on page 6, line 38, “may runs” should be “may run”

** on page 7, line 47, “In total, 239 total articles were” should be “In total, 239 articles were”

I would also like to add a recommendation for future works.

The authors clearly explain the reasons why they have selected only articles whose full text is in English.

This is an inherent limitation which does not affect the quality or the interest of the present work. However, an additional study (to be published independently, as a new work) considering articles in other languages for which the authors may get access, would be useful for the sake of completeness. I am not suggesting translators, because, for a study of this type, an author has to have direct contact with the original texts of the articles to be analysed. On the other hand, I tend to assume that if a pool of six Europe-based authors combines their knowledge of other languages, it will likely cover at least French and Spanish, what would mean covering additional (not-in-English) sources from French-speaking Africa and from most of Latin America (besides France and Spain); and German-language literature appears to be naturally accessible to most of the authors. This geographic scope entails adequate basis for a new informative work. Its publication would also encourage other researchers to undertake similar studies for literature in other languages. The importance of the topic would recommend an extension of the research to all existing sources, without excluding any language.

Decision letter (RSOS-211032.R0)

Dear Dr Ross-Hellauer

On behalf of the Editors, we are pleased to inform you that your Manuscript RSOS-211032 "Dynamics of Cumulative Advantage and Threats to Equity in Open Science: A Scoping Review" has been accepted for publication in Royal Society Open Science subject to minor revision in accordance with the referees' reports. Please find the referees' comments along with any feedback from the Editors below my signature.

Please submit your revised manuscript and required files (see below) no later than 7 days from today's (ie 06-Oct-2021) date. Note: the ScholarOne system will 'lock' if submission of the revision is attempted 7 or more days after the deadline. If you do not think you will be able to meet this deadline please contact the editorial office immediately.

on behalf of Professor Vania Zuin (Associate Editor) and Nick Pearce (Subject Editor)
 openscience@royalsociety.org

**Associate Editor Comments to Author (Professor Vania Zuin):
 Comments to the Author:**

Rev 1:

In "Dynamics of Cumulative Advantage and Threats to Equity in Open Science: A Scoping Review" the authors present a set of criticisms or points that have the potential to exacerbate inequalities in the scholarly community. Some of them are very valid and will be important for readership to be aware of. Some of the criticisms are difficult for me to comprehend. The authors wisely do not attempt to weigh the relative costs and benefits of OS. However, they take that mentality a bit too far in my opinion, and the reader is left to wonder what to do with the information provided. Is awareness of the potential problem sufficient? In some cases yes, but in most cases there should be some recommendation to revise how knowledge generation is valued, or how policies should be updated to provide broader guidance for researchers in various regions or disciplines.

I believe that I am part of the target audience of this paper, as I work closely with policy makers on open science initiatives (as a matter of personal policy, I disclose my name and role in the provided fields). Acknowledging my perception, my goal with this review is not to serve as a gatekeeper, as I think that criticisms should be widely shared. But I mostly want to help the authors reach me in order to provide a persuasive set of recommendations or lessons to be applied. That may result in a rather reductive view of the utility or purpose of this scoping review, but it nonetheless is what I would need to see in order to have it affect my opinion, which I think it will be capable of doing.

Specific questions or concerns are raised below, after quotes from the text.

The authors state: "In light of this, we should take seriously a strand of critique which links Open Science to broader trends to reshape the academy under neoliberal principles to emphasise market principles of competition, foregrounding its economic role in training the workforce and fostering new products and services, at the expense of the social mission to provide upward mobility for marginalised populations"

I don't follow how OS exacerbates this trend. The trend itself is clear, and it is clear how those forces could use OS, but it is not clear whether there is any feature of OS that makes it more or less susceptible to the trend. I know the authors are not trying to give a weighing of the costs and benefits of OS, but a contrast in this section (between factors that are more susceptible to these pressures under an OS regime than a "traditional" regime) would clarify to the readers (both those who are skeptical of the authors claims and those who are merely trying to follow their logical) how the authors are pointing out the particular risks associated with OS.

The Open Access section does not appear to add anything particularly new. It does synthesize the arguments, but the tension between OA and the cost that transfers to knowledge creators from knowledge consumers is fairly well known. As with my previous acknowledgement that the authors are not (and should not) attempt to weigh the relative costs and benefits, there is nonetheless a missing opportunity to point to the for-profit publishing organizations as the primary forces that are exacerbating the problem, and that no matter who pays (authors or readers), the winners are the publishers and the losers are those who are least able to afford to

publish or read. OS in general seems to have little overall effect, just a shift that publishers are responding to.

Open data. "...access is not enough to guarantee the utility of open data because the latter requires resources such as skills, money, and computing power [148]. Those working in environments where these are in short supply can fear that opening data sets will put them at a disadvantage" I don't follow the logic of this argument, which seems to be one of the two major points of section 4.3 (the other being that open data is primarily modeled after quantitative, STEM research that does not rely on human subjects). It is true that access to data is not sufficient to recognize its utility, but it at least seems necessary. If skills, money, and computing power are in low supply, then opaque data or open data will both result in lack of some result. This almost implies that, since not everyone can use some data, then no one should be able to. Perhaps it does increase inequality by virtue of the fact two groups not conducting research are equal, compared to two groups where one conducts research and the other doesn't are not equal. But if that is the conclusion, then I would recommend that the authors be explicit about it and then discuss its implications. A recommendation from that could be that "equality" should not be used as a primary argument for OS, as it could motivate an unintended race to the bottom.

Section 4.4 "...first movers will potentially have an advantage." I think an important point would be to articulate how first-movers and the existing overly privileged knowledge producers overlap. I can see some readers would assume that "first movers" is synonymous with "privileged," however, I think many who identify as first movers would also identify as smaller players who are less privileged than their peers.

"or further devalue qualitative approaches which cannot meet such criteria." I think this is an important point that the authors should emphasize more. I agree that OS is created with a particular sub-set of methodologies and values them over others.

"The term Open Science itself is often seen as exclusionary of the arts and humanities in the anglophone community. More pernicious, though, is the potential devaluation of epistemic diversity that attends Open Science's focus on transparency and reproducibility." In order to address this concern, the OS community could either a) be more inclusive of the arts and humanities as knowledge producers or b) assert that many of the recommendations are designed specifically to improve how quantitative, experimental inferences are made. If "a" is pursued, it seems to make "b" more difficult, because one cannot apply the same methods or expectations to both (as the authors rightly point out).

Rev. 2:

The researchers conducted a search of published and grey literature on the research area from January 2000 to the present, published in English, which yielded 3170 total results. Their research looking for kind of evidence and discourse which exist in the literature about the ways in which dynamics and structures of inequality could persist or be exacerbated in the transition to Open Science, across disciplines, regions and demographics, may contribute a lot ensuring in finding inequities in certain publications.

The researchers presented their goal in a clear way, used an adequate methodology, and composed the discussion and conclusions in a thorough way. In addition, they referred to the limitations of the study, following by a recommendation that "any potential for Open Science to actually drive inequalities must be taken seriously by the scientific community in order to realize the aim of making science truly open, collaborative and meritocratic."

Based on the above, I recommend accepting the manuscript.

Rev. 3:

he research topic is crucial for the scientific community.

The article is well written and highly informative. The information is presented in a balanced manner, which leaves full space for the reader to come to his/her independent conclusions. At the same time, the way in which it is presented stimulates reflections and thoughts, what is the naturally-expected outcome of this type of research.

I am convinced that this article needs to be made available to the scientific community as soon as possible.

Given that I do not find any faults requiring specific addressing, I strongly recommend that the article is published as it is.

I have two minor suggestions:

** on page 6, line 38, "may runs" should be "may run"

** on page 7, line 47, "In total, 239 total articles were" should be "In total, 239 articles were"

I would also like to add a recommendation for future works.

The authors clearly explain the reasons why they have selected only articles whose full text is in English.

This is an inherent limitation which does not affect the quality or the interest of the present work. However, an additional study (to be published independently, as a new work) considering articles in other languages for which the authors may get access, would be useful for the sake of completeness. I am not suggesting translators, because, for a study of this type, an author has to have direct contact with the original texts of the articles to be analysed. On the other hand, I tend to assume that if a pool of six Europe-based authors combines their knowledge of other languages, it will likely cover at least French and Spanish, what would mean covering additional (not-in-English) sources from French-speaking Africa and from most of Latin America (besides France and Spain); and German-language literature appears to be naturally accessible to most of the authors. This geographic scope entails adequate basis for a new informative work. Its publication would also encourage other researchers to undertake similar studies for literature in other languages. The importance of the topic would recommend an extension of the research to all existing sources, without excluding any language.

Reviewer comments to Author:

Reviewer: 1

Comments to the Author(s)

In "Dynamics of Cumulative Advantage and Threats to Equity in Open Science: A Scoping Review" the authors present a set of criticisms or points that have the potential to exacerbate inequalities in the scholarly community. Some of them are very valid and will be important for readership to be aware of. Some of the criticisms are difficult for me to comprehend. The authors wisely do not attempt to weigh the relative costs and benefits of OS. However, they take that mentality a bit too far in my opinion, and the reader is left to wonder what to do with the information provided. Is awareness of the potential problem sufficient? In some cases yes, but in most cases there should be some recommendation to revise how knowledge generation is valued, or how policies should be updated to provide broader guidance for researchers in various regions or disciplines.

I believe that I am part of the target audience of this paper, as I work closely with policy makers on open science initiatives (as a matter of personal policy, I disclose my name and role in the provided fields). Acknowledging my perception, my goal with this review is not to serve as a gatekeeper, as I think that criticisms should be widely shared. But I mostly want to help the authors reach me in order to provide a persuasive set of recommendations or lessons to be applied. That may result in a rather reductive view of the utility or purpose of this scoping review, but it nonetheless is what I would need to see in order to have it affect my opinion, which I think it will be capable of doing.

Specific questions or concerns are raised below, after quotes from the text.

The authors state: "In light of this, we should take seriously a strand of critique which links Open Science to broader trends to reshape the academy under neoliberal principles to emphasise market principles of competition, foregrounding its economic role in training the workforce and fostering new products and services, at the expense of the social mission to provide upward mobility for marginalised populations"

I don't follow how OS exacerbates this trend. The trend itself is clear, and it is clear how those forces could use OS, but it is not clear whether there is any feature of OS that makes it more or less susceptible to the trend. I know the authors are not trying to give a weighing of the costs and benefits of OS, but a contrast in this section (between factors that are more susceptible to these pressures under an OS regime than a "traditional" regime) would clarify to the readers (both those who are skeptical of the authors claims and those who are merely trying to follow their logical) how the authors are pointing out the particular risks associated with OS.

The Open Access section does not appear to add anything particularly new. It does synthesize the arguments, but the tension between OA and the cost that transfers to knowledge creators from knowledge consumers is fairly well known. As with my previous acknowledgement that the authors are not (and should not) attempt to weigh the relative costs and benefits, there is nonetheless a missing opportunity to point to the for-profit publishing organizations as the primary forces that are exacerbating the problem, and that no matter who pays (authors or readers), the winners are the publishers and the losers are those who are least able to afford to publish or read. OS in general seems to have little overall effect, just a shift that publishers are responding to.

Open data. "...access is not enough to guarantee the utility of open data because the latter requires resources such as skills, money, and computing power [148]. Those working in environments where these are in short supply can fear that opening data sets will put them at a disadvantage" I don't follow the logic of this argument, which seems to be one of the two major points of section 4.3 (the other being that open data is primarily modeled after quantitative, STEM research that does not rely on human subjects). It is true that access to data is not sufficient to recognize its utility, but it at least seems necessary. If skills, money, and computing power are in low supply, then opaque data or open data will both result in lack of some result. This almost implies that, since not everyone can use some data, then no one should be able to. Perhaps it does increase inequality by virtue of the fact two groups not conducting research are equal, compared to two groups where one conducts research and the other doesn't are not equal. But if that is the conclusion, then I would recommend that the authors be explicit about it and then discuss its implications. A recommendation from that could be that "equality" should not be used as a primary argument for OS, as it could motivate an unintended race to the bottom.

Section 4.4 "...first movers will potentially have an advantage." I think an important point would be to articulate how first-movers and the existing overly privileged knowledge producers overlap. I can see some readers would assume that "first movers" is synonymous with "privileged," however, I think many who identify as first movers would also identify as smaller players who are less privileged than their peers.

"or further devalue qualitative approaches which cannot meet such criteria." I think this is an important point that the authors should emphasize more. I agree that OS is created with a particular sub-set of methodologies and values them over others.

"The term Open Science itself is often seen as exclusionary of the arts and humanities in the anglophone community. More pernicious, though, is the potential devaluation of epistemic

diversity that attends Open Science's focus on transparency and reproducibility." In order to address this concern, the OS community could either a) be more inclusive of the arts and humanities as knowledge producers or b) assert that many of the recommendations are designed specifically to improve how quantitative, experimental inferences are made. If "a" is pursued, it seems to make "b" more difficult, because one cannot apply the same methods or expectations to both (as the authors rightly point out).

Reviewer: 2

Comments to the Author(s)

The researchers conducted a search of published and grey literature on the research area from January 2000 to the present, published in English, which yielded 3170 total results. Their research looking for kind of evidence and discourse which exist in the literature about the ways in which dynamics and structures of inequality could persist or be exacerbated in the transition to Open Science, across disciplines, regions and demographics, may contribute a lot ensuring in finding inequities in certain publications.

The researchers presented their goal in a clear way, used an adequate methodology, and composed the discussion and conclusions in a thorough way. In addition, they referred to the limitations of the study, following by a recommendation that "any potential for Open Science to actually drive inequalities must be taken seriously by the scientific community in order to realize the aim of making science truly open, collaborative and meritocratic."

Based on the above, I recommend accepting the manuscript.

Reviewer: 3

Comments to the Author(s)

The research topic is crucial for the scientific community.

The article is well written and highly informative. The information is presented in a balanced manner, which leaves full space for the reader to come to his/her independent conclusions. At the same time, the way in which it is presented stimulates reflections and thoughts, what is the naturally-expected outcome of this type of research.

I am convinced that this article needs to be made available to the scientific community as soon as possible.

Given that I do not find any faults requiring specific addressing, I strongly recommend that the article is published as it is.

I have two minor suggestions:

** on page 6, line 38, "may runs" should be "may run"

** on page 7, line 47, "In total, 239 total articles were" should be "In total, 239 articles were"

I would also like to add a recommendation for future works.

The authors clearly explain the reasons why they have selected only articles whose full text is in English.

This is an inherent limitation which does not affect the quality or the interest of the present work. However, an additional study (to be published independently, as a new work) considering articles in other languages for which the authors may get access, would be useful for the sake of completeness. I am not suggesting translators, because, for a study of this type, an author has to have direct contact with the original texts of the articles to be analysed. On the other hand, I tend to assume that if a pool of six Europe-based authors combines their knowledge of other languages, it will likely cover at least French and Spanish, what would mean covering additional (not-in-English) sources from French-speaking Africa and from most of Latin America (besides France and Spain); and German-language literature appears to be naturally accessible to most of the authors. This geographic scope entails adequate basis for a new informative work. Its publication would also encourage other researchers to undertake similar studies for literature in

other languages. The importance of the topic would recommend an extension of the research to all existing sources, without excluding any language.

===PREPARING YOUR MANUSCRIPT===

===PREPARING YOUR REVISION IN SCHOLARONE===

-- If you have uploaded ESM files, please ensure you follow the guidance at <https://royalsociety.org/journals/authors/author-guidelines/#supplementary-material> to include a suitable title and informative caption. An example of appropriate titling and captioning may be found at [https://figshare.com/articles/Table_S2_from_Is_there_a_trade-off_between_peak_performance_and_performance_breadth_across_temperatures_for_aerobic_sc ope_in_teleost_fishes_/3843624](https://figshare.com/articles/Table_S2_from_Is_there_a_trade-off_between_peak_performance_and_performance_breadth_across_temperatures_for_aerobic_scope_in_teleost_fishes_/3843624).

Author's Response to Decision Letter for (RSOS-211032.R0)

See Appendix A.

RSOS-211032.R1 (Revision)

Review form: Reviewer 1

Is the manuscript scientifically sound in its present form?

Yes

Are the interpretations and conclusions justified by the results?

Yes

Is the language acceptable?

Yes

Do you have any ethical concerns with this paper?

No

Have you any concerns about statistical analyses in this paper?

No

Recommendation?

Accept as is

Comments to the Author(s)

The authors have addressed the points I made sufficiently- either by modest revisions to their original points or, more generally, by more clearly asserting why their original point should stand. In both types of cases, I see no other concerns.

Decision letter (RSOS-211032.R1)

Dear Dr Ross-Hellauer,

It is a pleasure to accept your manuscript entitled "Dynamics of Cumulative Advantage and Threats to Equity in Open Science: A Scoping Review" in its current form for publication in Royal Society Open Science. The comments of the reviewer(s) who reviewed your manuscript are included at the foot of this letter.

on behalf of Professor Vania Zuin (Associate Editor) and Nick Pearce (Subject Editor)
openscience@royalsociety.org

Reviewer comments to Author:

Reviewer: 1

Comments to the Author(s)

The authors have addressed the points I made sufficiently- either by modest revisions to their original points or, more generally, by more clearly asserting why their original point should stand. In both types of cases, I see no other concerns.

Appendix A

Tony Ross-Hellauer
Open and Reproducible Research Group
TU Graz and Know-Center
8010 Graz, Austria
19th October 2021

Dear Editors of Royal Society Open Science,

On behalf of my co-authors (who have contributed to, reviewed and approved the amendments), I hereby resubmit our manuscript RSOS-211032 "Dynamics of Cumulative Advantage and Threats to Equity in Open Science - A Scoping Review".

We thank all reviewers for their valuable and constructive comments. We have revised in accordance with their suggestions for minor changes. Below you can find our detailed responses to each of the three reviewers' points.

In addition, we would like to point out that, since we also received rich feedback on the preprint version of this article (via Twitter and Hypothesis), we have also made revisions to respond to these points. For transparency we hence here detail each of those points, as well as our detailed response and any description of revisions.

We hope that having addressed these reviewer comments in this fashion that you will agree that our manuscript is now ready for acceptance for publication in Royal Society Open Science, but of course are very ready to respond if you have lingering concerns.

We thank you again for your time and care in appraising our manuscript, and look forward to hearing from you soon. Please do not hesitate to get in touch if you require any further information.

Yours sincerely,

Tony Ross-Hellauer

tross@know-center.at

Responses to reviewers and descriptions of revisions

Responses to Reviewer 1

R1 comment: "In "Dynamics of Cumulative Advantage and Threats to Equity in Open Science: A Scoping Review" the authors present a set of criticisms or points that have the potential to exacerbate inequalities in the scholarly community. Some of them are very valid and will be important for readership to be aware of. Some of the criticisms are difficult for me to comprehend."

- Author response: We thank the reviewer for their time and their critical feedback – addressing these points adds greatly to our work.

R1 comment: "The authors wisely do not attempt to weigh the relative costs and benefits of OS. However, they take that mentality a bit too far in my opinion, and the reader is left to wonder what to

do with the information provided. Is awareness of the potential problem sufficient? In some cases yes, but in most cases there should be some recommendation to revise how knowledge generation is valued, or how policies should be updated to provide broader guidance for researchers in various regions or disciplines. I believe that I am part of the target audience of this paper, as I work closely with policy makers on open science initiatives (as a matter of personal policy, I disclose my name and role in the provided fields). Acknowledging my perception, my goal with this review is not to serve as a gatekeeper, as I think that criticisms should be widely shared. But I mostly want to help the authors reach me in order to provide a persuasive set of recommendations or lessons to be applied. That may result in a rather reductive view of the utility or purpose of this scoping review, but it nonetheless is what I would need to see in order to have it affect my opinion, which I think it will be capable of doing.”

- Author response: We thank the reviewer for their constructive remarks, however we believe we are fundamentally at odds with this critique. As reviewer 1 correctly points out, we do not attempt to weigh the costs and benefits of OS with our paper, but rather to synthesize evidence and point out how inequalities shape outcomes of the OS transition. As authors, we fear that taking the step from problem diagnosis to problem solution would involve unnecessarily streamlining what we hopefully showed to be quite vexed, complex issues which do not admit of straightforward solutions. This scoping review is the start of a much broader programme of research undertaken within the ON-MERRIT (<https://on-merrit.eu/>) project. This work aims to map what is already known about the issues. Within the project we have then been conducting primary research to more closely investigate these issues, and in our final stages will work with stakeholders in co-creation activities to formulate community-endorsed recommendations for potential corrective policy actions by researchers, funders and institutions. Hence, at this stage, we do not wish to take the strongly normative stance desired by the reviewer, of making explicit recommendations over and above the general statements about overarching issues that appear in the discussion. Our ongoing work to investigate these issues and create recommendations will of course be shared with the community in due course.

R1 comment: The authors state: “In light of this, we should take seriously a strand of critique which links Open Science to broader trends to reshape the academy under neoliberal principles to emphasise market principles of competition, foregrounding its economic role in training the workforce and fostering new products and services, at the expense of the social mission to provide upward mobility for marginalised populations”. I don’t follow how OS exacerbates this trend. The trend itself is clear, and it is clear how those forces could use OS, but it is not clear whether there is any feature of OS that makes it more or less susceptible to the trend. I know the authors are not trying to give a weighing of the costs and benefits of OS, but a contrast in this section (between factors that are more susceptible to these pressures under an OS regime than a “traditional” regime) would clarify to the readers (both those who are skeptical of the authors claims and those who are merely trying to follow their logical) how the authors are pointing out the particular risks associated with OS.”

- Author response: We do not wish to claim that OS exacerbates the trend towards neoliberalization of academia. Rather, the point (taken from the reviewed literature) amounts to OS being (merely) an expression of these broader trends, albeit with a technological edge. The mechanism, if there is one, would be the emphasis OS has put on making research outputs publicly available, a notion which seems dangerously close to the idea that “sharing is caring” found, in different forms, in other influential examples of the data economy. This is what P. Mirowski (2018) refers to as the “platformisation” of science which actually denotes two related processes: a) the colonization of the research landscape by for-profit tools and firms,

and b) the research process being subject to increasing division of labour (Reichmann 2021, preprint) wherein smaller and smaller chunks of the process are made objects of public scrutiny (e.g. open projects, open lab notebooks). On the plus side, this process opens up new working opportunities for postdoctoral researchers (e.g. in the role of a data steward), but it does seem to involve division and ultimately retention of research labour into smaller and smaller bits for easier outsourcing. We have slightly revised the relevant section to make this clearer.

R1 comment: “The Open Access section does not appear to add anything particularly new. It does synthesize the arguments, but the tension between OA and the cost that transfers to knowledge creators from knowledge consumers is fairly well known. As with my previous acknowledgement that the authors are not (and should not) attempt to weigh the relative costs and benefits, there is nonetheless a missing opportunity to point to the for-profit publishing organizations as the primary forces that are exacerbating the problem, and that no matter who pays (authors or readers), the winners are the publishers and the losers are those who are least able to afford to publish or read. OS in general seems to have little overall effect, just a shift that publishers are responding to.”

- Author response: We appreciate the point Reviewer 1 makes. However, we would argue that the point is orthogonal to our main theme which is inequities in academia and how they affect the OS transition, notwithstanding the fact that it seems that publishers were quite happy to switch their publication models – indeed this would not add anything significant to the discussion. Rather, our aim was to suggest that OA tends to exacerbate inequalities within academia, between men and women, between well-resourced and under-resourced institutions which seems to run orthogonally to various OA-publishing models (whether Gold-OA or Green-OA, that is). While it is true that for-profit publishers win in all cases as it stands, there is a sense in which current OA models put some academics at a considerable disadvantage (as reviewer 1 rightly points out) while putting others at an advantage. Our aim was merely to point out how this situation tends to reflect historical inequalities along the lines of gender, region, etc. However, we appreciate the point that the for-profit publishers play a role in exacerbating the problem. We agree, and have revised the MS slightly to make this point more clearly.

R1 comment: “Open data. “...access is not enough to guarantee the utility of open data because the latter requires resources such as skills, money, and computing power [148]. Those working in environments where these are in short supply can fear that opening data sets will put them at a disadvantage” I don’t follow the logic of this argument, which seems to be one of the two major points of section 4.3 (the other being that open data is primarily modeled after quantitative, STEM research that does not rely on human subjects). It is true that access to data is not sufficient to recognize its utility, but it at least seems necessary. If skills, money, and computing power are in low supply, then opaque data or open data will both result in lack of some result. This almost implies that, since not everyone can use some data, then no one should be able to. Perhaps it does increase inequality by virtue of the fact two groups not conducting research are equal, compared to two groups where one conducts research and the other doesn’t are not equal. But if that is the conclusion, then I would recommend that the authors be explicit about it and then discuss its implications. A recommendation from that could be that “equality” should not be used as a primary argument for OS, as it could motivate an unintended race to the bottom.”

- Author response: Many thanks for this insightful comment, which we took as an invitation to clarify and hopefully sharpen the argument presented in the above paragraph. R1 points out an important clarification that needs to be made here concerning the origins of disadvantage and whether they are ethically relevant. The argument we wanted to make turns on observing the distinction between data access and data reuse. Even when granting access to data, this will not remove (all) barriers to reuse, as the latter depends on a host of other factors as well, such as knowledge, monetary resources, and computing power. In this vein, we agree with R1 that access to data is necessary if insufficient FOR REUSE. This gap between access and reuse, we argue, is precisely where forms of discrimination and unfair (dis)advantage take hold. We have now attempted to clarify this point by speaking more succinctly about reuse instead of utility. In that sense, we do not argue that open data increases inequality but rather that existing inequalities affect reuse, and so open data by itself is not going to change that (even while we agree that open data has positive effects, these are hampered by existing inequalities).

R1 comment: "Section 4.4 "...first movers will potentially have an advantage." I think an important point would be to articulate how first-movers and the existing overly privileged knowledge producers overlap. I can see some readers would assume that "first movers" is synonymous with "privileged," however, I think many who identify as first movers would also identify as smaller players who are less privileged than their peers."

- Author response: Many thanks to reviewer 1 for pointing out an important nuance in the argument. Whether someone is privileged or not is usually relative to a specific context. In our view, the way Open Science has hitherto been implemented at the level of institutions has been extremely resource-intensive. Effectively, the resources necessary for OS implementation are often primarily available to well-resourced institutions which suggests that only these can be first-movers in the first place, i.e., all first movers are necessarily privileged in some sense but not vice versa. The situation is entirely different at the individual level where the adoption of Open Science practices exhibits aspects of a grassroots movement. There, first movers are often early career researchers without permanent positions who are not privileged relative to researchers within their institutions who have permanent positions (but may still be privileged when compared to their access to resources via their institutions). There is then a sense in which institutional first movers are "absolutely" privileged, but individual first movers are only "relatively" privileged. The key here is to differentiate institutional actors (who need to be well-resourced) and individual actors (who are usually relatively privileged but not absolutely privileged). Since it seems to cause confusion, we decided to omit the phrase "first movers".

R1 comment: "'or further devalue qualitative approaches which cannot meet such criteria." I think this is an important point that the authors should emphasize more. I agree that OS is created with a particular sub-set of methodologies and values them over others."

- Author response: Thanks to reviewer 1 for their comment. We agree that this could be stressed a little more here but also point out that the point is one of two central arguments in section 4.3 (preceding this one). We therefore decided to only expand on the point slightly in this instance to state that in the same way that the FAIR guiding principles have been designed primarily for quantitative data, the same is true for quantitative methods.

R1 comment: ““The term Open Science itself is often seen as exclusionary of the arts and humanities in the anglophone community. More pernicious, though, is the potential devaluation of epistemic diversity that attends Open Science’s focus on transparency and reproducibility.” In order to address this concern, the OS community could either a) be more inclusive of the arts and humanities as knowledge producers or b) assert that many of the recommendations are designed specifically to improve how quantitative, experimental inferences are made. If “a” is pursued, it seems to make “b” more difficult, because one cannot apply the same methods or expectations to both (as the authors rightly point out).”

- Author response: Many thanks for this comment. In fact, we do not see a lot of evidence for the OS community adopting a), and some evidence that the recommendations that do exist and are being institutionalized in the form of, e.g., institutional Open Science policies, do not attempt to be inclusive of SSH (Social Sciences and Humanities) disciplines, nor do they attempt to disclose their narrow epistemological focus. Rather, we fear that there is a real threat in Open Science identifying good science with whatever research conforms to the principles of reproducibility and transparency, effectively marginalizing SSH fields further. As already stated, it was not our aim here to make direct recommendations – this is a scoping review to compile and synthesise existing evidence. The project from which this MS stems is currently doing work with stakeholders to co-create recommendations, which will be shared at a later stage.

Responses to Reviewer 2

R2 comment: ““The researchers conducted a search of published and grey literature on the research area from January 2000 to the present, published in English, which yielded 3170 total results. Their research looking for kind of evidence and discourse which exist in the literature about the ways in which dynamics and structures of inequality could persist or be exacerbated in the transition to Open Science, across disciplines, regions and demographics, may contribute a lot ensuring in finding inequities in certain publications. The researchers presented their goal in a clear way, used an adequate methodology, and composed the discussion and conclusions in a thorough way. In addition, they referred to the limitations of the study, following by a recommendation that “any potential for Open Science to actually drive inequalities must be taken seriously by the scientific community in order to realize the aim of making science truly open, collaborative and meritocratic.” Based on the above, I recommend accepting the manuscript.”

- Author response: We thank the reviewer for their time and their positive feedback.

Responses to Reviewer 3

R3 comment: “The research topic is crucial for the scientific community. The article is well written and highly informative. The information is presented in a balanced manner, which leaves full space for the reader to come to his/her independent conclusions. At the same time, the way in which it is presented stimulates reflections and thoughts, what is the naturally-expected outcome of this type of research. I am convinced that this article needs to be made available to the scientific community as soon as possible. Given that I do not find any faults requiring specific addressing, I strongly recommend that the article is published as it is.”

- Author response: We thank the reviewer for their time and their positive feedback.

R3 comment: “** on page 6, line 38, “may runs” should be “may run””

- Author response: Many thanks to reviewer 3, we have fixed the mistake.

R3 comment: “** on page 7, line 47, “In total, 239 total articles were” should be “In total, 239 articles were””

- Author response: Many thanks, mistake has been fixed.

R3 comment: *“I would also like to add a recommendation for future works. The authors clearly explain the reasons why they have selected only articles whose full text is in English. This is an inherent limitation which does not affect the quality or the interest of the present work. However, an additional study (to be published independently, as a new work) considering articles in other languages for which the authors may get access, would be useful for the sake of completeness. I am not suggesting translators, because, for a study of this type, an author has to have direct contact with the original texts of the articles to be analysed. On the other hand, I tend to assume that if a pool of six Europe-based authors combines their knowledge of other languages, it will likely cover at least French and Spanish, what would mean covering additional (not-in-English) sources from French-speaking Africa and from most of Latin America (besides France and Spain); and German-language literature appears to be naturally accessible to most of the authors. This geographic scope entails adequate basis for a new informative work. Its publication would also encourage other researchers to undertake similar studies for literature in other languages. The importance of the topic would recommend an extension of the research to all existing sources, without excluding any language.”*

- Author response: We thank the reviewer for this excellent suggestion, which we have added to our final section

Responses to other feedback received via Twitter and Hypothesis

Twitter

<https://twitter.com/CameronNeylon/status/1413759649002754048>

Cameron Neylon: “Don’t know if you’re still editing, but one thing I’d add under FAIR data is the problematising of data as a concept itself across different epistemic systems (with indigenous knowledge as a strong example). Came up a bunch in @OCSNet work and a bit in my IDRC study+ // The concept of data as “facts removable from context” that underpins the arguments for FAIR and Open Data needs to be unpicked and the conceptualisation is strongly linked to risks for expropriation of indigenous and other deeply contextual knowledges”

- Author response: Many thanks, this is an excellent suggestion with which we wholeheartedly agree. In fact, there is now a vast discussion on the contextuality of data (e.g. Borgman 2012, 2015); earlier and more recent work into disciplinary data practices (the scope of this work is too far to appreciate it in full in the short paragraph though) suggests that data and their meaning always necessarily depend on their context of origin; in fact, Sabina Leonelli suggests that the work of data curators is precisely that of decontextualizing datasets (as best as possible) which is a precondition for reuse. We have added text to the first paragraph of section 4.3 attempting to succinctly explain the idea of data contextuality.

Hypothesis annotated comments on SocArXiv preprint version:

<https://osf.io/preprints/socarxiv/d5fz7/>

[Please note that due to a bug (which was reported to Hypothesis but has not yet been fixed), some annotations from another preprint (which predate publication of our preprint) appear as Hypothesis annotations on our preprint - we obviously do not include those comments here]

Egonw 11 Jul Public

Key threats include

How does your method distinguish between risks resulting from the Open Science ideology and resulting from how people implement/do Open Science? Since the "creative" use of terms like Open Science (e.g. by Elsevier and other publishers) and Open Access (by basically everyone), the threats of implementations can be huge, but so is the variation. The abstract, however, does not seem to distinguish between them, giving the impression that these threats are 1. unique to Open Science, 2. intrinsic to Open Science. Neither of these are supported by this study, I think.

- Author response: Many thanks for pointing out what is indeed a key distinction, but we note this comment is made in response to the abstract only. However, beyond the abstract, we feel this issue is handled in depth throughout the manuscript.

Egonw 11 Jul Public

new risks of bias and exclusion

New compared to what? How does your method compare with that "what"?

- Author response: New in the sense of a reconstructed Open Science environment creating new opportunities for exclusion, with the transition and wide-spread adoption of Open science could put groups at a disadvantage that are already (traditionally) marginalized. This is what we hoped to show with our scoping study.

Lakens 11 Jul Public

Open Science has been proposed at least in part as a corrective for some of these issues

By who? Where? Can you provide references? Regardless who said this, there must be many people who disagree with such a broad definition of open science - open science is typically used to refer to science that is open - not science that is equitable. Equitable science is an orthogonal goal - important, but unrelated to Open Science.

Author response: We believe the 2-3 paragraphs which follow this statement demonstrate the point that increased equity is often included as a key goal of Open Science (giving references). Regarding the point that not everyone would agree with this - we explicitly state that equity is "one aim of Open Science amongst others", which again refers to the fact that there are broader definitions of OS (that we endorse) and narrower ones. We also discuss how the ambiguous aims of Open Science amongst a multitude of actors tend to blur this picture. There is therefore an important sense in which equity goes to the heart of Open Science (under a broad definition of OS), and it is this view of OS that we endorse in our work. In fact, showing how equity is precisely not orthogonal is one of the aims of the scoping review which

(hopefully) demonstrates how the adoption of Open Science relates to many instances of (in)equity.

Lakens 11 Jul Public

equity is one aim of Open Science amongst other

It would be useful to distinguish schools - there are people who do not believe in making 'Open Science' a container concept that means everything.

- Author response: We believe this point is made very clearly within the existing draft – viz., reference to Fecher and Friesike’s five schools, the point about the essential ambiguity of the term Open Science, the point raised in the discussion that the fact that no unified definition emerged which allowed it in some ways to mean all things to all people ... A similar point can be found in our response made above to the charge of misconstruing the scope of Open Science (in relation to equity). While we do subscribe to a broad reading of the term “Open Science” we certainly do not think it should be a container concept; again following Fecher and Friesike, we suggest it’s more of an umbrella term (whether this is a good thing or not is another matter, but this is not the point of our work).

Egonw 11 Jul Public

Open Access

Can you clarify what you mean with Open Access here? What definition are you using? Does this include the use of "Open Access" by publishers, which does not always include the rights to modify and redistribute the science?

- Author response: This was indeed a bit unclear, thank you for the clarification. We have now added a definition of what we mean by OA – “OA, whereby scientific publications are distributed online, free of cost or other access barriers under open licensing conditions”.

Egonw 11 Jul Public

Open/FAIR Data

Please clarify what you mean with Open/FAIR Data. Is this Open **AND** FAIR Data, or Open **OR** FAIR Data? Because FAIR Data and Open Data are different concepts, it sounds important to me to clarify this.

- Author response: We did wish to discuss potential inequalities implicated in both Open Data and FAIR Data (and are fully aware that the two are different). We have now separated the two more explicitly throughout the text by removing the backslash and instead speaking of “Open Data and FAIR data” or “Open Data or FAIR Data” where appropriate. The respective section 4.3 on Open and FAIR data has also been revised to reflect this better, even while there, there was a clearer division of the two to begin with.

Egonw 11 Jul Public

above all by Open Science advocates

Studies that look into the changes in risks of doing Open Science compared to doing Closed Science is very much welcome to me.

- Author response: Us too. The point of the scoping review was to lay out what evidence on risks associated with the OS transition is already there.

Egonw 11 Jul Public

Methods

I note that the study looks into literature selectively, with keywords around open topics (with the exception of a few like "justice" and "FAIR data" which are about closed science too). I find it hard to see how this corpus allows you to draw conclusions on threats resulting from doing Open Science if you do not compare this to the threats of doing Closed Science. Are the threats new? How did they change? Are they bigger or smaller than with Closed Science. Can you clarify?

- Author response: Any literature review is necessarily selective – however, our search strategy (clearly defined) aimed to identify threats to equity associated with open science. As such, using keywords defining open science topics in conjunction with words like justice, inequity, etc. seems to us an entirely sensible approach. The comparison to “closed science” comes from the individual studies which are assessed.

Egonw 11 Jul Public

ScopusandWeb of Science

Do these capture research literature from countries outside the South well? What effect does that have on the conclusions regarding Open Science policies outside the North?

- Author response: This is covered in the limitations and future work section at the end of the MS

Egonw 11 Jul Public

resultsTOPIC

Open Source has been an early driver of Open Science, going back to the late nineties. Why is it not included?

- Author response: Open source was not included as a term for reasons of precision – including it returned far too many studies which were not relevant and would have made the screening process unmanageable (simply put, of the vast literature on Open Source and equity, very few are relevant to Open Science per se). However, we did address Open Source and exclusion in Open Science within our section on “open methods and open infrastructure”.

Egonw 11 Jul Public

R "FAIR Data"

FAIR data is not Open Data. Including articles about FAIR data pulls in a lot of articles about closed/proprietary data.

- Author response: See earlier comment/revisions regarding the relationship between Open and FAIR data.

Egonw 11 Jul Public

has been driven by richer, global North nations

How did you ensure you captured policies from South America and Asia? Your search uses English-literature oriented search solutions? How did you make sure these do not introduce a bias against Open Science policies in the South?

- Author response: Potential exclusion of manuscripts from the global south is indeed a limitation of our study. We therefore covered this point in the limitations and future work section at the end of the revised MS.

Egonw 11 Jul Public

APC-based OA hence risks stratifications

How does this relate to hybrid journals often charging more APC than full CC-BY journals?

- Author response: We believe this point is orthogonal here since our aim was to qualify potential inequities but not necessarily quantify them.

Lakens 11 Jul Public

undermine their utility

This is a very strong statement - In monetary terms, it is important to note that APC for people in some countries is 0 after a waiver, and in other countries it costs research budget. A Strong statement requires a much better cost-benefit analysis here to be believable.

- Author response: Our point here was merely that waivers, as currently implemented, are not (for the reasons given) a great response to barriers to publication caused by APCs. We weakened the statement to make it less directly normative, and more descriptive of the criticisms others have made (“mean that they are often criticised as a weak response to an urgent issue [103,104].”)

Lakens 11 Jul Public

If so, the landscape of open data may not be the democratisation of knowledge, but just a further mechanism whereby the rich get richer

This paragraph is not convincing. The claim that open data is primarily accessible to institutes with more budget might be true - but this is irrelevant. You need to argue that the increase in accessibility has lower marginal utility for poorer institutes - but the opposite must be true. There is too much confirmation bias in this paragraph - you need to do a better job providing a fair cost-benefit analysis - it reads as if you were searching for arguments to support a preconceived notion - not like you tried to honestly weigh costs and benefits.

- Author response: Many thanks for this feedback. The comment makes a point which is fair given the inadvertent identification of “reuse” with “utility” we committed in the paragraph. However, we do not agree with the indictment that the paragraph should be about marginal utility at all. Reuse requires resources in a way that speaking of mere utility does not. The point is not that institutes with more resources have easier access to open data – that is false almost by virtue of the definition of “open data” which are open to anyone – but rather whether an institution can make effective use of openly accessible data. There, the ethnographic research we cite does seem to show that effective reuse (not access!) depends on the availability of a host of other resources such as enough money to pay for data analysts, for instance, where we would expect a clear disadvantage in the Global South. These resources are subject to an

unfair distribution which the open data rhetoric rarely touches upon (and nor should it, probably). Even if anybody can access a given data set, not everybody will be in a position to reuse it (assuming everything else is equal, e.g. knowledge, background, research interest), and the moderating variable (so to speak) is the available resources. We have attempted to clarify the entire paragraph and bring out the central argument better which is that even while open data have positive effects, existing inequalities (such as endowment with resources) moderate these positive effects. Given that data reuse depends on resources such as computing power (which determines, e.g., how many analyses can be run), there is a clear sense in which existing disadvantages in terms of resource distribution moderate how well nominally open data can be reused.